# ViscoReg: Neural Signed Distance Functions via Viscosity Solutions

**Meenakshi Krishnan**                                                    *mkrishn9@umd.edu*
*Perceptual Intelligence and Reality Lab (PIRL)*
*Dept. of Mathematics*
*University of Maryland, College Park*

**Ramani Duraiswami**                                                     *ramanid@umd.edu*
*Perceptual Intelligence and Reality Lab (PIRL),*
*UMIACS & Dept. of Computer Science,*
*University of Maryland, College Park*

**Reviewed on OpenReview:** *https://openreview.net/forum?id=DWnMkBU4sF*

## Abstract

Implicit Neural Representations (INRs) that learn Signed Distance Functions (SDFs) from point cloud data represent the state-of-the-art for geometrically accurate 3D scene reconstruction. However, training these Neural SDFs involves enforcing the Eikonal equation, an ill-posed equation that also leads to unstable gradient flows. Numerical Eikonal solvers have relied on viscosity approaches for regularization and stability. Motivated by this well-established theory, we introduce ViscoReg, a novel regularizer for Neural SDF methods, and theoretically prove that it stabilizes training. Empirically, ViscoReg outperforms state-of-the-art approaches such as SIREN, DiGS, StEik, and HotSpot across most metrics on ShapeNet, the Surface Reconstruction Benchmark, 3D scene reconstruction and reconstruction from real scans. We also establish novel generalization error estimates for Neural SDFs in terms of the training error, using the theory of viscosity solutions. Our empirical and theoretical results provide confidence in the general applicability of our method.

## 1 Introduction

Implicit neural representations (INRs) encode continuous signals, such as images, sounds, 3D surfaces, or scenes (Mildenhall et al., 2021; Park et al., 2019; Mescheder et al., 2019). Neural networks mapping input coordinates to signal values are used as compact, high-resolution representations of the underlying signal. Neural Signed Distance Functions (SDFs) (Park et al., 2019) extend this approach to 3D scene reconstruction. The model learns a function that maps spatial coordinates to their signed distance from a surface manifold, implicitly defining the surface as the zero level set of the function. They are trained on the input point cloud data by constraining the signed distance to be zero on the surface, and optionally, using surface normal information. In the absence of normal information, previous methods suffer a severe degradation in reconstruction quality.

While normals may be precomputed from the input point cloud, this is expensive and typically error-prone. Without normals, constraining the network to be zero on the surface can lead the network to collapse to the trivial zero function during training. Enforcing the Eikonal partial differential equation (PDE):

$$\|\nabla u(x)\|_2 = 1 \text{ for } x \in \Omega, \ u(x) = 0 \text{ for } x \in \partial\Omega, \tag{1}$$

via the *Eikonal loss* ensures that the network learns a valid SDF (Gropp et al., 2020). Here, $\Omega$ is a bounded domain, and $\partial\Omega$ is the sufficiently smooth boundary surface we aim to reconstruct. However, the Eikonal loss alone may not be enough for good reconstruction (Ben-Shabat et al., 2022), and it presents two fundamental challenges. First, training with this regularizer can cause instabilities, leading the network to converge to

suboptimal local minima with large errors, as has been demonstrated both theoretically and empirically (Yang et al., 2023). Second, the Eikonal equation is inherently ill-posed, with multiple solutions satisfying 1 almost everywhere (Lipschitz solutions, see Sec. 3). *Viscosity solutions* represent the physically meaningful solution to the Eikonal equation (and to the broader class of Hamilton-Jacobi equations (Crandall & Lions, 1983)). The SDF is the unique viscosity solution of the Eikonal. This leads to an important question for Neural SDFs: **With infinitely many solutions to the Eikonal equation, why is minimizing the PDE residual loss on a finite training set sufficient to ensure convergence to the unique viscosity solution (i.e., the SDF)?**

We use the theory of viscosity solutions to approach both challenges discussed above. To address the theoretical ill-posedness, we rigorously establish bounds on the INR generalization error using properties of viscosity solutions, and classical PDE inequalities. To the best of our knowledge, *this is the first work to provide bounds on the global error between the learned function and the ground truth SDF in terms of the training error.* To address the practical instability, we consider the well-posed parabolic equation, which adds a viscosity/diffusion term to the Eikonal:

$$\|\nabla u_\varepsilon\|_2 = 1 + \varepsilon \Delta u_\varepsilon. \tag{2}$$

The viscosity solution $u$ of equation 1 is recovered in the limit $\varepsilon \to 0$ of $u_\varepsilon$. The vanishing viscosity method is important in the analysis of Hamilton-Jacobi class of equations, and care is taken in classical numerical analysis to arrive at the viscosity solution rather than one of the infinitely many other Lipschitz solutions. For instance, the Fast Marching method (Sethian, 1999), a widely used Eikonal solver, computes viscosity solutions via level-set techniques. Motivated by these, we propose **a novel regularization technique** that incorporates a dynamically scaled viscous term into the Eikonal loss during training. This is proven theoretically to stabilize training, and empirically improves reconstruction quality, while avoiding the pitfalls of other regularizations that either lack physical rigor (e.g., constraining the SDF to be harmonic (Ben-Shabat et al., 2022)), require normals (Atzmon & Lipman, 2020b), or overfit to input noise (Yang et al., 2023).

Together, our theoretical and empirical results are both founded in the theory of viscosity solutions. The SDF is the unique viscosity solution of the Eikonal equation, whose zero level set is exactly the surface we seek to reconstruct. The classical vanishing viscosity method, which recovers this solution as the limit of parabolic equations, motivates ViscoReg, which we establish provably stabilizes the Eikonal training dynamics. Our main contributions can be summarized as follows:

• Generalization error bounds are presented showing that the learned solution converges to the unique viscosity solution as the training error and quadrature error tend to zero.

• Neural SDFs, which are currently state-of-the-art for surface reconstruction, rely on the Eikonal constraint which is ill-posed and has unstable training dynamics. This can lead to convergence to suboptimal local minima and poor reconstruction. We propose *ViscoReg*, a novel training regime for Neural SDF based on the vanishing viscosity method with a dynamically scaled loss. We justify this by analyzing the gradient flow of its variational formulation, and demonstrate its ability to stabilize training for high-frequency components. Our loss maintains the same hyperparameter count as DiGS (Ben-Shabat et al., 2022), StEik (Yang et al., 2023) or HotSpot (Wang et al., 2025).

• We compare our work with current SOTA methods on several reconstruction benchmarks to demonstrate significant improvements.

## 2 Related Work

### 2.1 Surface Reconstruction

Reconstructing surfaces from point clouds is a long-studied problem in computer vision that is challenging due to non-uniform point sampling, noisy normal estimations, missing surface regions, and other data imperfections (Berger et al., 2017). The problem is highly ill-posed, as there are multiple surfaces that can fit a finite set of points (Sulzer et al., 2024). Traditional methods include triangulation (Cazals & Giesen, 2006), Voronoi diagrams (Amenta et al., 1998), and alpha shapes (Bernardini et al., 2002). Implicit function methods using radial basis functions (Carr et al., 2001) and Poisson surface reconstruction (Kazhdan et al., 2006) are also well-studied. More recent non-neural approaches include Neural Splines (Williams et al., 2021), which use kernel formulations arising from infinitely wide shallow networks, and Shape As Points (Peng

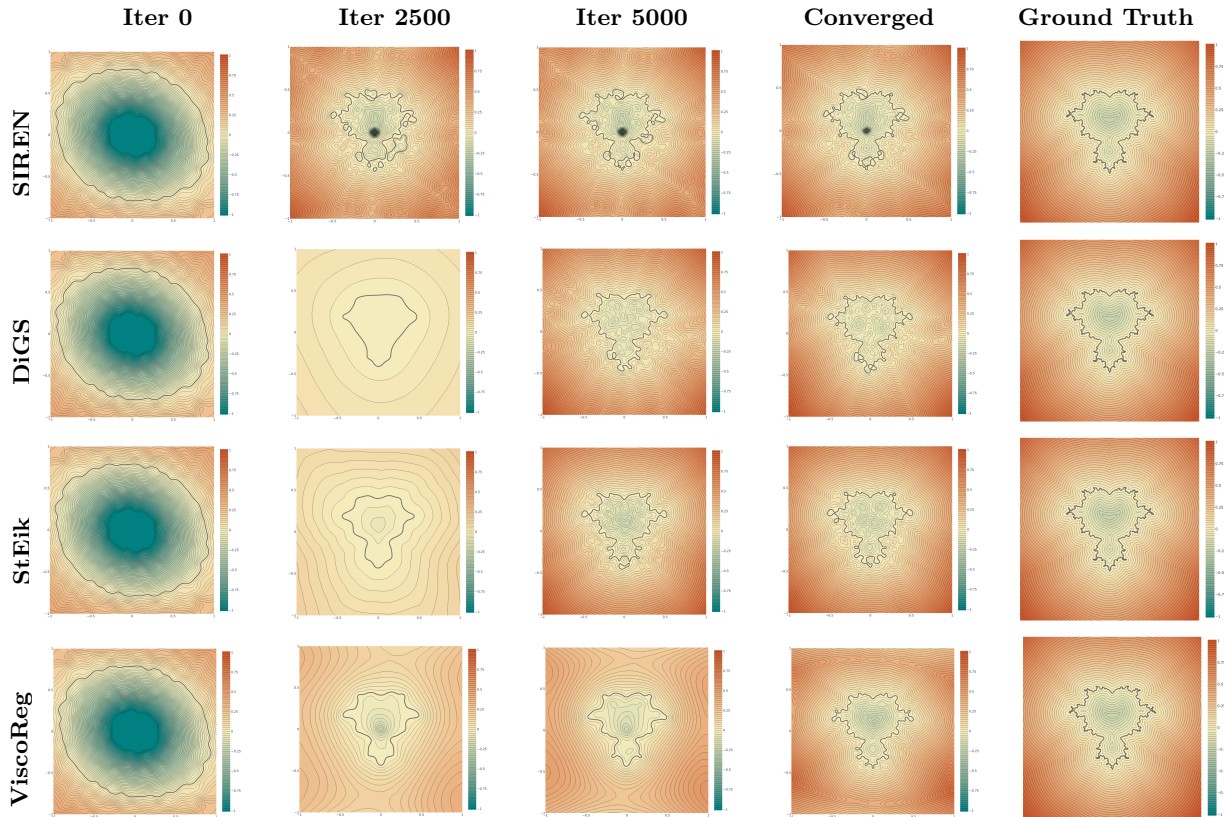

Figure 1: Reconstructing the 2D fractal Mandelbrot set using different Neural SDF techniques. **SIREN:** Converges quickly but the boundary is poorly reconstructed with many self-intersections. **DiGS:** Overly smoothed boundary in early iterations, with the final reconstructed boundary being disconnected and self-intersecting. **StEik:** While it avoids oversmoothing, it struggles with spurious self-intersections, disconnections and not capturing fine detail. **ViscoReg:** Smoothly converges to the underlying complex boundary, maintaining its intricate structure throughout training.

et al., 2021), which represents surfaces using a differentiable Poisson solver. Methods based on differentiable 3D Gaussian splatting (Kerbl et al., 2023) have also been increasingly employed for this task (Guédon & Lepetit, 2024; Krishnan et al., 2025; Waczyńska et al., 2024). Relevant to our work is VisCo Grids (Pumarola et al., 2022), a grid-based method incorporating viscosity; however, it is not a neural network-based approach and uses a fixed, non-decaying viscosity coefficient estimated from the grid resolution, unlike our method. While viscosity-based regularization has a long history in numerical Eikonal solvers (Sethian, 1999), modern reconstruction pipelines rely almost exclusively on neural implicit representations trained by gradient-based optimization. This shift raises a fundamental question on how viscosity methods can improve the training dynamics of neural SDFs, which we address in this work.

## 2.2 Implicit Neural Representations

INRs are a popular approach in volumetric representation due to their high resolution and compactness (Cao & Taketomi, 2024; Chen & Zhang, 2019; Lombardi et al., 2019; Ma et al., 2020; Michalkiewicz et al., 2019; Mildenhall et al., 2021; Müller et al., 2022; Sitzmann et al., 2019a;b; Wang et al., 2024). They have demonstrated success in encoding shapes by learning SDFs or occupancy functions (Mescheder et al., 2019). DeepSDF (Park et al., 2019) was the first to learn an SDF with a neural network, but relied on ground truth SDFs for supervision, which are usually unavailable. SAL (Atzmon & Lipman, 2020a) proposed learning the SDF directly from point cloud data, constraining the function to be zero on the surface. SALD (Atzmon & Lipman, 2020b) added normal supervision; IGR (implicit geometric regularization) (Gropp et al., 2020) introduced the Eikonal loss to ensure that the learned function is a valid SDF. PHASE (Lipman, 2021)

proposed a density function that converges to an occupancy function. While PHASE used viscosity theory to justify the convergence of their occupancy representation, our work establishes the first generalization error bounds for standard Neural SDFs. SIREN (Sitzmann et al., 2020) used a sine activation function, which allows computation of higher-order derivatives, such as the Laplacian term in this work.

DiGS (Ben-Shabat et al., 2022) minimizes the Laplacian of the learned function, showing improved performance without normals. However, the SDF Laplacian corresponds to the mean curvature of the surface, and its minimization can lead to over-smoothing of fine detail (see Sec. 5). StEik (Yang et al., 2023) identified training instabilities with the Eikonal loss and proposed a directional divergence regularizer, similar to the gradient-Hessian alignment constraint in Wang et al. (2023a). However, this is a direct mathematical consequence of the Eikonal and naturally holds when this constraint is satisfied. Empirically, as seen in StEik, it overfits noise in the input. More recently, HotSpot (Wang et al., 2025) addresses the stability of Neural SDFs by proposing a loss function derived from a screened Poisson equation. This contrasts with our approach, which is grounded in the classical PDE theory of viscosity solutions to directly regularize the Eikonal equation.

### 2.3 Neural PDE solvers

PDEs are foundational models in applications like computer graphics and wave propagation. While traditionally solved with numerical methods like finite difference and finite element (LeVeque, 1992; Ames, 2014), neural networks are increasingly used to approximate PDE solutions (Han et al., 2018; Blechschmidt & Ernst, 2021; Sirignano & Spiliopoulos, 2018). Notably, Physics-Informed Neural Networks (PINN) introduced by Karniadakis et al. (2021), incorporate PDE residuals and boundary conditions in the loss. The Neural SDF method is a specific PINN for the Eikonal equation. Despite their empirical success, learning theory for these solvers is still nascent. Generalization theory aims to understand how well the network generalizes to unseen data given the training error. Results have been established for neural methods for PDE in abstract settings (Shin et al., 2020; Mishra & Molinaro, 2022; Chen et al., 2025), and for specific PDEs (De Ryck & Mishra, 2022; Hu et al., 2021; Berner et al., 2020; Zubov et al., 2021). We extend this analysis to Neural SDFs, providing intuition on why the network should converge to the correct solution and establishing bounds on the worst-case deviation from the ground truth.

## 3 Error Analysis

We present novel theoretical results on generalization error bounds for Neural SDF, starting with a brief overview of Neural SDFs and viscosity solutions. This also motivates the ViscoReg training regime introduced in Sec. 4. The primary motivation behind the study in this section is to establish theoretical bounds on how well the Neural SDF method approximates the unique viscosity solution. In our work, $\Omega \subset \mathbb{R}^3$ represents an open, connected, bounded domain with sufficiently smooth boundary $\partial\Omega$. Lebesgue and Sobolev spaces are represented as $L^p(\Omega)$ and $W^{k,p}(\Omega)$, equipped with standard norms, $\|\cdot\|_{L^p(\Omega)}$, (denoted for simplicity as $\|\cdot\|_p$) and $\|\cdot\|_{W^{k,p}(\Omega)}$ for $1 \le k, p \le \infty$ (definitions in appendix). The space of continuous functions on $\Omega$ is denoted as $C(\Omega)$ with the $L^\infty(\Omega)$ norm, and $C^k(\Omega)$ is the space of $k$-times differentiable functions with $C^k$ norm.

### 3.1 Neural Signed Distance Functions

A Neural SDF $u_\theta : \Omega \to \mathbb{R}$ is a network, parametrized by weights $\theta \in \mathbb{R}^d$, approximating an SDF whose zero level set is $\partial\Omega$. Since ground-truth SDF values for non-manifold points are not usually available, training is supervised using the *manifold constraint* $\mathcal{L}_m$ and the *non-manifold penalization constraint* $\mathcal{L}_{nm}$. These ensure that $u_\theta$ is zero on the manifold, and non-zero away from it.

$$\mathcal{L}_m(u_\theta) = \int_{\partial\Omega} |u_\theta(x)| \ dx, \quad \mathcal{L}_{nm}(u_\theta) = \int_{\Omega\setminus\partial\Omega} e^{-\alpha|u_\theta(x)|} \ dx. \tag{3}$$

Additionally, the *Eikonal constraint* $\mathcal{L}_{eik}$ that specifies the norm of the gradient to be one is enforced.

$$\mathcal{L}_{eik}(u_\theta) = \int_\Omega \|\|\nabla u_\theta\|_2 - 1\|^p \ dx \text{ for } p = 1, 2. \tag{4}$$

The combined loss with hyperparameters $\alpha_m$, $\alpha_{nm}$, $\alpha_e$ and $\alpha >> 1$ is:

$$\mathcal{L}(u_\theta) = \alpha_m \mathcal{L}_m(u_\theta) + \alpha_{nm} \mathcal{L}_{nm}(u_\theta) + \alpha_e \mathcal{L}_{eik}(u_\theta). \tag{5}$$

We do not consider a normal loss as normals may need to be obtained via error-prone pre-processing. The input is the surface point cloud $\mathcal{P}_{\partial\Omega} := \{x_i\}_{i=1}^N \subset \partial\Omega$, and uniformly sampled non-manifold points from the domain $\mathcal{P}_\Omega := \{y_j\}_{j=1}^M \subset \Omega$. The integrals of equation 3-4 are discretized as:

$$L_m(u_\theta; \mathcal{P}_{\partial\Omega}) = \frac{1}{N} \sum_{i=1}^N \|u_\theta(x_i)\|_p, \ x_i \in \mathcal{P}_{\partial\Omega},$$

with $L_{nm}$ and $L_{eik}$ defined analogously for $\mathcal{L}_{nm}$, $\mathcal{L}_{eik}$. Thus, the optimization problem is:

$$\arg\min_{\theta \in \mathbb{R}^d} \left(\alpha_m L_m(u_\theta; \mathcal{P}_{\partial\Omega}) + \alpha_e L_{eik}(u_\theta; \mathcal{P}_{\partial\Omega} \cup \mathcal{P}_\Omega) + \alpha_{nm} L_{nm}(u_\theta; \mathcal{P}_\Omega)\right), \tag{6}$$

where $u_\theta \in \mathcal{F}_{\text{NN}}$ , where $\mathcal{F}_{\text{NN}}$ is the class of fully connected SIREN networks, with weights $\theta \in \mathbb{R}^d$.

## 3.2 Generalization Error

A considerable challenge for Eikonal equations equation 1 is the lack of uniqueness - there exist infinitely many continuous functions that satisfy the equation almost everywhere (referred to as Lipschitz solutions), i.e., outside of a set of measure zero. Consider the one-dimensional Eikonal equation $\|u'(x)\|_2 = 1$, with boundary conditions $u(0) = u(1) = 0$ in $[0, 1]$. Any zig-zag function with slopes $\pm 1$ satisfying the boundary conditions is a Lipschitz solution (the points with $C^1$ discontinuities are a set of measure 0), whereas the SDF solution is $u(x) := \min(x, 1-x)$. In many applications, the meaningful solution is the *viscosity solution*, introduced by Crandall & Lions (1983). These solutions possess maximum and stability properties, which make the analysis of Eikonal-and more broadly, of Hamilton-Jacobi equations—more tractable. Viscosity solutions inherit these properties from the solutions of the well-posed parabolic equations (2), which they approximate in the limit (Calder, 2018).

The SDF is the unique viscosity solution of the Eikonal equation 1; and hence the viscosity solution's zero level set exactly represents the surface geometry. Thus, we want to quantify how well the Neural SDF can approximate the viscosity solution. Using properties of the viscosity solutions, and classical inequalities in PDE theory, we provide a novel generalization error estimate. The estimate is provided when the $L^1$ norm is used for the Eikonal loss. It is extended to the $L^2$ case in the appendix.

The computational domain is often chosen as a bounding box tightly fitted to the surface, enclosing the shape. For analysis, we consider the domain to be the volume enclosed by the surface. Since the trained network will not exactly satisfy the Eikonal equation 1, consider the more general formulation of the boundary value problem (BVP):

$$\|\nabla u(x)\|_2 = f(x), \ x \in \Omega, \quad u(x) = g(x), \ x \in \partial\Omega. \tag{7}$$

where $f \in C^\infty(\bar{\Omega})$, $g \in C(\partial\Omega)$, for $\bar{\Omega} = \Omega \cup \partial\Omega$. When $f \not\equiv 1$, $u$ is not the SDF, but rather the shortest arrival time of a wavefront propagating from $x \in \bar{\Omega}$ to $\partial\Omega$. The function $f$ represents the "slowness" (reciprocal of the speed) in the medium, while $g$ acts as an exit-time penalty.

Denote by $\text{USC}(\bar{\Omega})$ and $\text{LSC}(\bar{\Omega})$, the space of upper and lower semi-continuous functions, respectively. The viscosity solution $u \in C(\bar{\Omega})$ of the Eikonal equation is defined below.

**Definition 3.1 (Viscosity Solution)** *A function $u \in \text{USC}(\bar{\Omega})$ is a* viscosity subsolution *of (7) if for all $x_0 \in \bar{\Omega}$ and all $\phi \in C^\infty(\mathbb{R}^3)$ such that $u - \phi$ has a local maximum at $x_0$, we have:*

$$\begin{cases} \|\nabla\phi(x_0)\|_2 - f(x_0) \leq 0, & \text{if } x_0 \in \Omega, \\ \min\{\|\nabla\phi(x_0)\|_2 - f(x_0), u(x_0) - g(x_0)\} \leq 0, & \text{if } x_0 \in \partial\Omega. \end{cases}$$

*Similarly, $u \in \text{LSC}(\bar{\Omega})$ is a* viscosity supersolution *of equation 7 if for all $x_0 \in \bar{\Omega}$ and all $\phi \in C^\infty(\mathbb{R}^3)$ such that $u - \phi$ has a local minimum at $x_0$, the following inequality holds:*

$$\begin{cases} \|\nabla\phi(x_0)\|_2 - f(x_0) \geq 0, & \text{if } x_0 \in \Omega, \\ \max\{\|\nabla\phi(x_0)\|_2 - f(x_0), u(x_0) - g(x_0)\} \geq 0, & \text{if } x_0 \in \partial\Omega. \end{cases}$$

*Then, $u \in C(\Omega)$ is a* viscosity solution *of (7) if it is both a viscosity subsolution and a supersolution.*

To obtain the required bounds, we establish a few preliminary results for these viscosity solutions.

**Lemma 1** *Let $u_1, u_2 \in C(\bar{\Omega})$ be viscosity solutions of the Eikonal equation $\|\nabla u\|_2 = f$, subject to the respective boundary conditions $u_{1|\partial\Omega} = g_1$, $u_{2|\partial\Omega} = g_2$, for $g_1, g_2 \in C(\partial\Omega)$. Then:*

$$\|u_1 - u_2\|_\infty \le \|g_1 - g_2\|_\infty. \tag{8}$$

Lemma 1 shows that equation 7 has at most one continuous viscosity solution. Next, we provide a stability estimate that shows the sensitivity of the viscosity solution to the slowness function.

**Lemma 2** *Let $u_1$, $u_2$ be unique viscosity solutions of $\|\nabla u\|_2 = f_1$, $\|\nabla u\|_2 = f_2$, respectively, with $u_{1|\partial\Omega} = u_{2|\partial\Omega} = 0$. Here, $f_1$, $f_2 \in C^\infty(\mathbb{R}^3)$, and assume, $\exists\, C_f > 0$ such that $0 < \frac{1}{C_f} \le f_1, f_2 < C_f$. Then the solutions satisfy:*

$$\|u_1 - u_2\|_\infty \le C_\Omega C_f^{-2}\|f_1 - f_2\|_\infty, \tag{9}$$

*where $C_\Omega$ is a constant corresponding to the diameter of $\Omega$.*

The proofs of both lemmas are in the appendix (Crandall et al., 1984; Calder, 2018). Now, let $\theta^* \in \mathbb{R}^d$ be the minimizer of the optimization (6) obtained via gradient descent algorithms, and let $u_{\theta^*} \in C^\infty(\bar{\Omega})$ be the corresponding network. Note that $u_{\theta^*}$ is smooth, since we use the sine activation function in SIREN. To analyze the error of the network $u_\theta^*$, which only approximately satisfies the Eikonal equation, we must consider it as an exact solution to a perturbed Eikonal equation, where the PDE residual corresponds to the slowness function $f_{\theta^*} \in C^\infty(\bar{\Omega})$ and the boundary error becomes the boundary condition $g_{\theta^*} \in C^\infty(\partial\Omega)$:

$$\|\nabla u_{\theta^*}(x)\|_2 = f_{\theta^*}(x),\ x \in \Omega, \quad u_{\theta^*}(x) = g_{\theta^*}(x),\ x \in \partial\Omega. \tag{10}$$

We assume that the network $u_{\theta^*}(x)$ satisfies the following conditions.

*Assumption 1:* The gradient of $u_{\theta^*}$ is bounded away from zero. Specifically, for all $x \in \Omega$, we have $0 < \frac{1}{C_{\theta^*}} \le \|\nabla u_{\theta^*}(x)\|_2 \le C_{\theta^*}$, for $C_{\theta^*} > 0$.

If $\theta^*$ is a sufficiently good local minima, it is natural that Assumption 1 holds, since the ground-truth SDF $u$ satisfies $\|\nabla u(x)\|_2 = 1 > 0$, for a.e. $x \in \Omega$.

*Assumption 2:* The input point cloud is such that the discrete sum used to calculate the boundary and Eikonal loss is a sufficiently good approximation of the true continuous integral. Specifically, $\mathcal{P}_{\partial\Omega} = \{x_i\}_{i=1}^N$ satisfies the quadrature error bound:

$$\left| \int_{\partial\Omega} |g(x)|^p dx - \frac{1}{N}\sum_{i=1}^N |g(x_i)|^p \right| \le C_g N^{-\beta}, \tag{11}$$

for $g \in C^\infty(\partial\Omega)$, $p = 1, 2$, and $\beta > 0$. This assumption is quite general, essentially requiring that as $N \to \infty$, the sample $\ell^p$ norm converges to the true $L^p$ norm. In the case of uniform sampling, $\beta$ takes the value $1/3$. For Monte Carlo random sampling, $\beta = 1/2$ for sufficiently smooth functions (Mishra & Molinaro, 2023). Since the point cloud data may be obtained through sensors, we consider the more general $\beta$ to account for irregularities in the sampling process. Ignoring measurement errors, we consider the sampling process to be deterministic, while non-uniform.

We empirically verify that Assumption 1 holds for our trained networks in Table 5 in the appendix. We also refer the interested reader to Lin et al. (2024); Pais et al. (2024) for detailed studies on optimal sampling strategies for neural implicit representations.

Since $\Omega$ is bounded, and the network has bounded weights, $\|u_\theta\|_{C^k(\Omega)} \le C_k < \infty$ for all finite $k$. This also implies that the network (and its derivatives) is bounded in $W^{k,p}$ norm for finite $k$ and $1 \le p \le \infty$. Denote $\|f_{\theta^*}\|_{W^{6,1}(\Omega)}, \|g_{\theta^*}\|_{W^{6,1}(\partial\Omega)} \le M_{\theta^*}$ for $M_{\theta^*} > 0$. Here, the choice of $k = 6$ is determined by the requirements of the interpolation inequality used in the proof of Theorem 1.

This brings us to the main theoretical results of our paper.

**Theorem 1** *Suppose Assumptions 1-2 hold. Consider the minimizer $\theta^* \in \mathbb{R}^d$ of (6) and let $u_{\theta^*} \in C^\infty(\bar{\Omega})$ be the network parametrized by $\theta^*$. Let $u \in C(\bar{\Omega})$ be the solution to equation 1. Then, the generalization error is bounded as:*

$$\|u - u_{\theta^*}\|_\infty \lesssim M_{\theta^*}(L_m(u_{\theta^*}))^{\frac{1}{2}} + M_{\theta^*} C_{\theta^*}^{-2} (L_{eik}(u_{\theta^*}))^{\frac{1}{2}} + \mathcal{O}((M+N)^{-1/6}) + \mathcal{O}(N^{-\frac{\beta}{2}}). \quad (12)$$

*The constants in $\lesssim$ depend only on $\bar{\Omega}$.*

**Proof Sketch:** We first decompose $\|u - u_{\theta^*}\|_\infty$, into terms controlled by the boundary error ($g_{\theta^*}$) and by the PDE residual ($f_{\theta^*}$). We apply stability estimates from Lemmas 1 and 2 to bound these terms. To connect the continuous, worst-case bounds to discrete training losses, the Gagliardo-Nirenberg interpolation inequality is used to relate $L^\infty$ norms to $L^1$ norms. Finally, we bound the $L^1$ norms using discrete sample losses, $L_m$ and $L_{eik}$, yielding the final result after accounting for the quadrature error using Assumption 2. The full proof is in the appendix.

At first glance, the generalization bound may seem expected, as it suggests that small training error leads to better generalization. However, this result is non-trivial in the context of PDE solutions, where there is no fundamental reason why minimizing the PDE residual and boundary loss at finitely many points should lead the network to converge to a solution of the continuous formulation of a nonlinear PDE. This is particularly insightful for the ill-posed Eikonal equation, which admits infinitely many solutions, only one of which is the viscosity solution (the true SDF). This *provides a theoretical guarantee* that the learned function is close to the viscosity solution in the $L^\infty$-sense, provided the training and quadrature errors are low. It also offers insight into how training error controls the worst-case deviation from the correct viscosity solution.

In surface reconstruction, the metric of interest is the quality of the zero-level set mesh (extracted using the Marching Cubes algorithm) compared to a ground-truth mesh. This is primarily evaluated using mesh quality metrics such as Intersection over Union (IoU) to measure volumetric overlap, and Chamfer distance to measure the average nearest-neighbor distance between the two discrete surfaces. A small $L^\infty$ error restricts the zero-level set of the learned function to remain close to the true surface, directly controlling the mesh quality metrics.

## 4 ViscoReg

### 4.1 Energy Formulations and Gradient Flow

An important problem in many applications is to find a function $u : \Omega \subset \mathbb{R}^n \to \mathbb{R}$ that minimizes a functional $E(u)$, representing an energy/loss function. The gradient flow defines the continuous evolution of $u$ along the path of steepest descent for $E(u)$. It may be obtained in the continuum limit of the gradient descent method for the minimization problem, and is given by:

$$u_t = -\nabla E(u). \quad (13)$$

Here, $t$ is an artificial time parameter (the continuous limit of discrete iterations of gradient descent), and $\nabla E(u)$ represents the Fréchet derivative of $E$ with respect to $u$. When $u$ is restricted to neural networks $u_\theta$ parametrized by weights $\theta \in \mathbb{R}^d$, the optimization is performed in the finite-dimensional parameter space. The resulting update to the function $u_\theta$ is understood as a projection of the ideal, unconstrained gradient flow onto the tangent space spanned by the network's basis functions (Yang et al., 2023). As the network's representational capacity increases, this basis more closely approximates the full function space, and the projected gradient flow approximates the unconstrained equation 13. Hence, we study the unconstrained gradient flow to provide insight into the training process. Computing the Fréchet derivative of the loss functional $\mathcal{L}_{eik}(u)$ (equation 3), we see that the gradient flow closely resembles the heat equation with:

$$u_t = \nabla \cdot (g(\|\nabla u\|_2) \nabla u), \quad g(s) = \begin{cases} \frac{1}{s} - 1, & p = 2, \\ \text{sign}(s-1), & p = 1. \end{cases}$$

Observe that $g$ can be positive or negative making the above equation a Forward-Backward heat equation. The backward nature, however, destabilizes the PDE. The gradient flow of the Eikonal loss has been studied

by Yang et al. (2023), who propose a stabilizing directional divergence regularizer, but as shown in Sec. 5, there is room for improvement. They show that adding a Laplacian energy term (as in Ben-Shabat et al. (2022)) can also stabilize training. However, since the SDF Laplacian is the mean curvature on the surface, it should not be minimized in areas of fine detail.

## 4.2 ViscoReg

Beyond providing generalization estimates, the viscosity framework also suggests a practical training strategy to deal with the unstable Eikonal gradient flow. The viscosity solution possesses an important property: it can be recovered as the limit of solutions to well-posed parabolic equations (see Figure 5 in the appendix for an illustration).

**Theorem 2 (Crandall & Lions (1983))** *For each $\varepsilon > 0$, let $u_\varepsilon \in C^2(\bar{\Omega}) \cap C(\bar{\Omega})$ denote the unique solution to equation 2. Then $u_\varepsilon \to u$ uniformly, as $\varepsilon \to 0^+$, where $u$ is the unique viscosity solution of (1).*

Motivated by Theorem 2, to stabilize the Eikonal gradient flow, we propose adding a decaying viscosity term:

$$\mathcal{L}(u_\theta) = \alpha_m \mathcal{L}_m(u_\theta) + \alpha_{nm} \mathcal{L}_{nm}(u_\theta) + \alpha_v \mathcal{L}_{veik}(u_\theta). \tag{14}$$

Here, $\mathcal{L}_{veik}$ represents the viscous Eikonal loss that we refer to as *ViscoReg* given by:

$$\mathcal{L}_{veik}(u_\theta) = \int_\Omega |\|\nabla u_\theta(x)\|_2 - 1 - \varepsilon\Delta u_\theta|^p \ dx, \quad p = 1, \ 2, \tag{15}$$

where $\varepsilon > 0$ is a hyperparameter decayed to zero over the course of training. Note that this is different from the DiGS loss because we are not minimizing the divergence with this regularization. The main motivation behind this regularization comes from Theorem 2 which states that the viscosity solution to the Eikonal (in Definition 3.1) is a limit of solutions to parabolic equation 2.

The added viscosity term lends stability to the Eikonal loss formulation. Let $r(u) = \|\nabla u\|_2 - 1 - \varepsilon\Delta u$ denote the viscous residual. For $p = 1$, computing the Fréchet derivative of $\mathcal{L}_{veik}$ gives the gradient flow equation:

$$u_t = \nabla \cdot \left( \text{sign}(r) \frac{\nabla u}{\|\nabla u\|_2} \right) + \varepsilon\Delta(\text{sign}(r)). \tag{16}$$

For analytical ease, we replace $\text{sign}(r)$ function with a smooth approximation (such as the sigmoid function) $S(r)$ such that $S(0) = 0$ and $S'(0) = c > 0$. Linearising the resulting non-linear PDE around its stationary solution $u_0 = \mathbf{a} \cdot x$ (with $\|\mathbf{a}\|_2 = 1$, say $\mathbf{a} = [1, 0, 0]^T$), we get:

$$u_t = c(\mathbf{a} \cdot \nabla)^2 u - c\varepsilon^2 \Delta^2 u. \tag{17}$$

Taking the Fourier transform of the above PDE gives:

$$\hat{u}_t = -c[(\mathbf{a} \cdot \omega)^2 + \varepsilon^2 \|\omega\|_2^4]\hat{u} \implies \hat{u} = \exp\left(-ct[(\mathbf{a} \cdot \omega)^2 + \varepsilon^2 \|\omega\|_2^4]\right). \tag{18}$$

The Fourier symbol $\lambda(\omega) = -c[(\mathbf{a} \cdot \omega)^2 + \varepsilon^2 \|\omega\|_2^4]$ is non-positive for all wavevectors since $c > 0$. This implies that as $t \to \infty$, we have $\hat{u} \to 0$ for all frequency modes, and that the equation is unconditionally stable.

Similar results are presented for the case $p = 2$ in the appendix. Thus, enforcing the viscous Eikonal PDE over the inviscid version in the initial phases of training, not only encourages convergence to the physically meaningful solution, but also can be theoretically proven to stabilize the Eikonal training. As proof of concept, we demonstrate the boundary reconstruction of a complex Mandelbrot fractal with different methods in Fig. 1. DiGS without normals, results in an overly smoothed boundary during early iterations. After the annealing phase, where the divergence weight is set to zero, the reconstructed boundary becomes self-intersecting and disconnected. Other state-of-the-art methods are also plagued with similar challenges. In contrast, ViscoReg smoothly converges to the highly curved boundary, maintaining its intricate structure throughout the process.

Note that as $\epsilon$ is annealed to zero well before training concludes, the final phase of training optimizes the standard inviscid Eikonal loss. Consequently, the generalization bound of Theorem 1 applies directly to the ViscoReg solution, with the viscous term serving only to stabilize the initial optimization trajectory.

## 5 Results

**Implementation Details:** We evaluate the proposed regularization term on different surface reconstruction tasks, specifically, the Surface Reconstruction Benchmark (Berger et al., 2013), a scene reconstruction task from Sitzmann et al. (2019b), ShapeNet (Chang et al., 2015) and real scans from Huang et al. (2024). Meshes are extracted using the Marching cubes algorithm (Lorensen & Cline, 1998) using a grid with shortest axis 512 tightly fitted onto the surface. We use the sine activation function proposed in SIREN to compute the second derivatives needed for our task. For all our experiments, we find a simple piecewise linear decay of $\varepsilon$ to be sufficient. We note that most state-of-the-art methods (including DiGS, StEik, and HotSpot) use annealing for their loss terms; ViscoReg introduces no additional hyperparameters beyond those already present in these baselines. Further implementation details are listed in the appendix.

Our main point of comparison involves SoTA methods such as DiGS (Ben-Shabat et al., 2022), StEik (Yang et al., 2023), HotSpot (Wang et al., 2025), Neural Singular Hessian (Wang et al., 2023b) and NeurCADRecon (Dong et al., 2024). However, note that StEik introduces two key techniques to achieve its results: (1) directional divergence regularizer, (2) quadratic layers in the network architecture. Our work introduces a theoretically motivated regularizer. So, besides the reported results, for an apples-to-apples comparison between the two methods, we report results on (a) StEik's regularizer with standard linear layers and the same architecture as our method and (b) StEik's architecture with our regularizer. Unless specified, all qualitative and quantitative results are presented for ViscoReg with linear layers, and StEik with quadratic layers. Methods marked "quad" correspond to the quadratic architecture. As in DiGS and related works, we evaluate our methods on the Chamfer distance metric ($d_C$), and the Hausdorff distance ($d_H$) metric for the Surface Reconstruction Benchmark. For the ShapeNet dataset, we report the squared Chamfer distance and the Intersection over Union (IoU) between the reconstructed shapes and ground truth. For reconstruction from real scans, we report Chamfer Distance, F-score and Normal Consistency Score (NCS).

Table 1: Results on SRB. $d_C$ : Chamfer and $d_H$: Hausdorff distance. $\Delta d_C$, $\Delta d_H$ denote mean deviation from the best method.

| Method | $d_C \downarrow$ | $d_H \downarrow$ | $\Delta d_C$ | $\Delta d_H$ |
|---|---|---|---|---|
| IGR wo n | 1.38 | 16.30 | 1.2 | 13.61 |
| SIREN wo n | 0.42 | 7.67 | 0.23 | 4.98 |
| SAL | 0.36 | 7.47 | 0.18 | 4.78 |
| IGR+FF | 0.96 | 11.06 | 0.78 | 8.37 |
| PHASE+FF | 0.22 | 4.96 | 0.04 | 2.27 |
| AGH (W23) | 0.19 | 2.98 | 0.01 | 0.29 |
| NSH | 0.20 | 3.73 | 0.02 | 1.04 |
| NeurCADRecon | 0.20 | 3.70 | 0.02 | 1.01 |
| VisCo Grids wo n | 0.34 | 4.39 | 0.16 | 1.95 |
| HotSpot | 0.19 | 3.17 | 0.01 | 0.48 |
| DiGS | 0.19 | 3.52 | 0.0 | 0.73 |
| StEik (linear) | 0.20 | 4.56 | 0.02 | 1.87 |
| Ours | **0.18** | **2.76** | 0.0 | 0.07 |
| StEik (quad) | **0.18** | 2.80 | 0.0 | 0.11 |
| Ours (quad) | **0.18** | **2.69** | 0.0 | 0.0 |

Table 2: Ablation on $\varepsilon$ decay for mean Chamfer and Hausdorff metrics in SRB. BL$\times x$= baseline scaled by $x$.

| Method | $d_C \downarrow$ | $d_H \downarrow$ |
|---|---|---|
| BL | **0.18** | 2.76 |
| BL $\times 2$ | 0.18 | 3.17 |
| BL $\times 0.5$ | 0.19 | 5.06 |
| Fast decay (0 @ 20%) | 0.19 | 3.51 |
| Slow decay(0 @ 90%) | 0.18 | 3.28 |
| Piecewise Const. | 0.20 | 5.17 |
| Quintic | 0.19 | 3.89 |
| $\varepsilon = 0$ (SIREN [47]) | 0.42 | 7.67 |
| SoTA best (HotSpot) | **0.19** | **3.17** |

**Results on the Surface Reconstruction Benchmark (SRB):** SRB consists of five noisy shapes as point clouds with normals. For fair evaluation we compare our method with the normal-free versions of SoTA. To train the network we used 5 hidden layers and 128 neurons. For $\varepsilon$, we used an annealing strategy, setting $\varepsilon = 0.5$ initially and decaying to zero through piece-wise linear schedule. This decay schedule does not add extra hyperparameters because a similar annealing strategy was used for the divergence terms in DiGS and StEik. We used the MFGI initialization from DiGS. Results for the Chamfer and Hausdorff distances between ground truth meshes are in Table 1. Our method improves upon all other methods in this task. There is considerable improvement in the Hausdorff distance, even though we use approximately 50% fewer parameters than DiGS or SIREN.

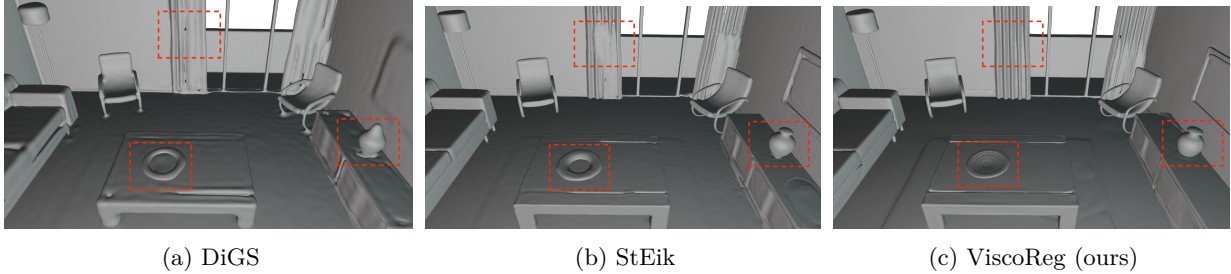

(a) DiGS              (b) StEik              (c) ViscoReg (ours)

Figure 2: Results from the scene reconstruction benchmark from Sitzmann et al. (2019b). The DiGS mesh (a) is missing fine details like the sofa legs, accurate vase shape on the right, and picture frame details. StEik (b) performs better but struggles with fine details such as the curtains and plate on the table. The ViscoReg mesh (c) reconstructs fine details with high fidelity.

*Viscosity parameter decay ablation:* Baseline decay for all shapes is initial $\varepsilon = 0.5$, decayed linearly at 20/40/60/80% iterations to 0.4/0.04/0.005/0 for ViscoReg (linear). See Table 2 for ablation. Many "reasonable" decays work well; an optimal schedule may be obtained via coarse grid search. When the baseline is reduced by half, the performance degrades and is close to the "no-viscosity" case (i.e. SIREN). This validates the necessity for viscosity stabilization of the Eikonal. Ablation decay schedules are provided in the appendix. We also provide additional ablation on the decay schedule for surface reconstruction from real scans in the appendix.

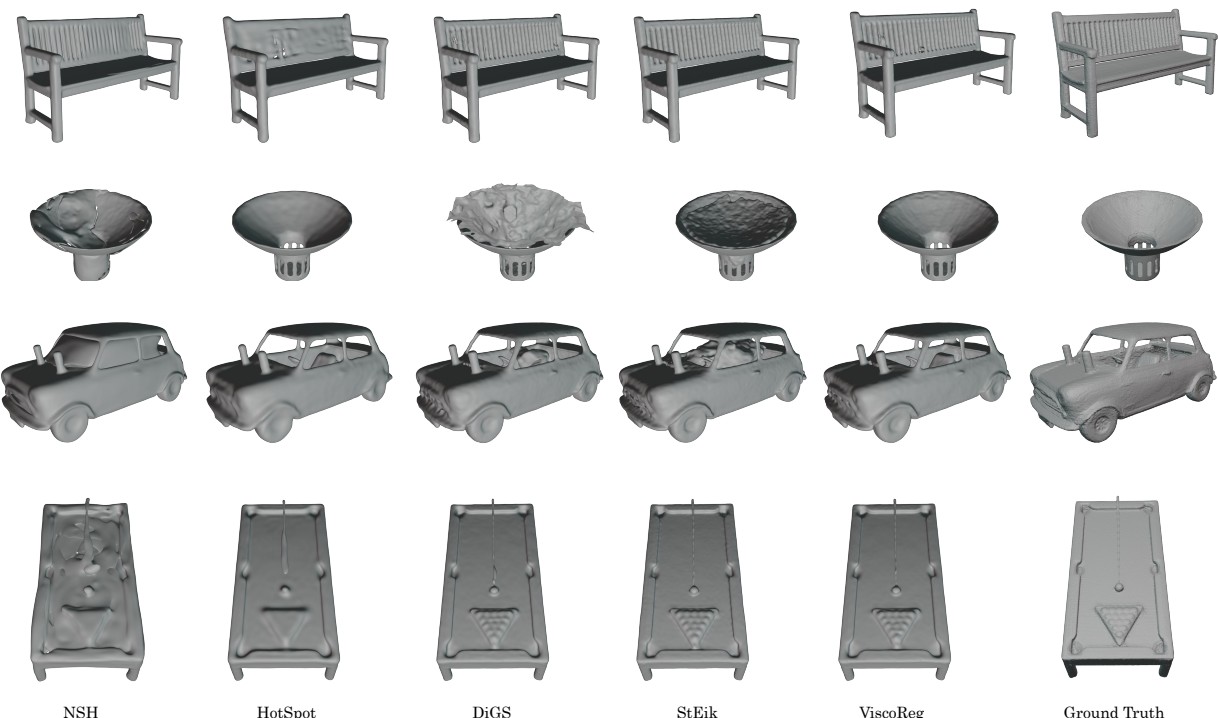

NSH          HotSpot          DiGS          StEik          ViscoReg          Ground Truth

Figure 3: Results from ShapeNet for examples from bench, lamp, car, and table (top to bottom) categories. NSH recovers lumpy surfaces and struggles with details. HotSpot causes tearing of the geometry (see bench and the pool table) and also causes lumps for simple shapes. DiGS and StEik results do not maintain sharp details, and exhibit ghost pieces and other artifacts. ViscoReg mesh avoids ghost geometry and reconstructs fine surface details with high fidelity. The results are from surfaces reconstructed with StEik (quadratic layers), and others with standard linear layers.

**Scene Reconstruction from Sitzmann et al. (2019b):** We use 8 layers and 512 neurons with 10M sample points as in the original dataset. Qualitative results are in Figure 2. Without normal information, methods like SIREN report ghost geometries (Ben-Shabat et al., 2022). Due to the smoothing effect of the Laplacian term, DiGS does not recover fine details such as sofa legs, vase and picture frames. StEik recovers details somewhat better but still struggles with more intricate detailing like picture frames, curtains, and the plate rim. Our method recovers the fine details reconstructing the surface with greater fidelity, even though we do not use normal information or quadratic layers.

**Surface Reconstruction from Real Scans:** To further demonstrate ViscoReg's robustness, we test it on the real scans benchmark of Huang et al. (2024), which consists of 20 noisy range scans of real objects. We report Chamfer Distance, F-score and Normal Consistency Score (NCS). For fair comparison, we test all methods with the same architecture (4 layers, 256 channels). We use the decay schedule starting at 1.0 linearly decaying to 0 by 60% iterations for all shapes. Despite VisCo Grids using normal information, ViscoReg outperforms all methods on 2 out of 3 metrics while

| Method | $d_C \times 10^{-2}$ ↓ | F-score ↑ | NCS ↑ |
|---|---|---|---|
| VisCo Grids w n | 32.11 | **88.52** | 94.20 |
| SIREN wo n | 52.35 | 75.88 | 90.36 |
| NeurCADRecon | 38.55 | 84.09 | 93.74 |
| NSH | 38.23 | 83.97 | 93.84 |
| HotSpot | 36.23 | 83.00 | 95.17 |
| DiGS | 37.49 | 86.11 | 94.72 |
| StEik (lin) | 38.45 | 86.31 | 94.73 |
| StEik (quad) | 39.56 | 82.62 | 89.82 |
| Ours (lin) | **30.55** | 86.50 | **95.67** |

Table 3: Comparisons on real scans (Huang et al., 2024). Only Visco Grids uses normals.

showing extremely competitive results on the other. Notably, we achieve the best results across metrics among methods not using normals. Qualitative results are presented in Figure 4 where ViscoReg excels at capturing fine details (eg., remote) while also maintaining shape integrity (inside of the prism flower pot).

Table 4: Our method is top 2 in every metric compared to SoTA, showing significant improvement in mean squared Chamfer distances. Top group (SPSR–DiGS +n) uses normal supervision; remaining methods do not. Bottom two have quadratic layers.

| method | Squared Chamfer ↓ | | | IoU ↑ | | |
|---|---|---|---|---|---|---|
| | mean | median | std | mean | median | std |
| SPSR | 2.22e-4 | 1.7e-4 | 1.76e-4 | 0.643 | 0.673 | 0.158 |
| IGR | 5.13e-4 | 1.13e-4 | 2.15e-3 | 0.810 | 0.848 | 0.152 |
| SIREN | 1.03e-4 | 5.28e-5 | 1.93e-4 | 0.827 | 0.910 | 0.233 |
| FFN | 9.12e-5 | 8.65e-5 | 3.36e-5 | 0.822 | 0.840 | 0.098 |
| NSP | 5.36e-5 | 4.06e-5 | 3.64e-5 | 0.897 | 0.923 | 0.087 |
| DiGS +n | 2.74e-4 | 2.32e-5 | 9.90e-4 | 0.920 | 0.977 | 0.199 |
| SIREN wo n | 3.08e-4 | 2.58e-4 | 3.26e-4 | 0.309 | 0.295 | 0.201 |
| SAL | 1.14e-3 | 2.11e-4 | 3.63e-3 | 0.403 | 0.394 | 0.272 |
| NSH | 7.60e-5 | 3.40e-5 | 1.49e-4 | 0.729 | 0.934 | 0.357 |
| HotSpot | **4.17e-5** | 2.35e-5 | 7.67e-4 | 0.936 | 0.976 | 0.158 |
| DiGS | 1.32e-4 | 2.55e-5 | 4.73e-4 | 0.939 | 0.974 | 0.126 |
| StEik (lin) | 1.36e-4 | 2.34e-5 | 9.34e-4 | **0.963** | **0.981** | 0.091 |
| ViscoReg (lin) | 5.27e-5 | **2.32e-5** | **1.08e-4** | 0.952 | 0.978 | **0.083** |
| StEik (quad) | 6.86e-5 | **6.33e-6** | 3.34e-4 | **0.967** | **0.984** | 0.088 |
| ViscoReg(quad) | **3.72e-5** | 2.17e-5 | **7.88e-5** | 0.959 | **0.984** | 0.086 |

**ShapeNet:** The dataset consists of 3D CAD models of a variety of object categories. Following Williams et al. (2021), we evaluate on 20 shapes per category across 13 categories. This preprocessing pipeline ensures consistent normal orientations and converts internal structures into manifold meshes. We use an architecture of 4 hidden layers and 256 channels as in Ben-Shabat et al. (2022). Results for HotSpot are also presented with our same 4/256 model.

The results in Table 4 and Figure 3 clearly demonstrate quantitative and qualitative improvements with respect to the SoTA. With linear layers, ViscoReg shows a 61% decrease in the mean squared Chamfer distance compared to StEik, while achieving comparable IoU scores (within 1% on mean, and 0.3% in median IoU). With quadratic layers, in the squared Chamfer metric, we achieve 45% reduction in mean error compared to StEik. This, combined with the lower variance of our results, indicates that our method avoids failure cases better in comparison to StEik. Thus, the proposed regularization improves reconstruction without smoothing out fine details and thin structures. More results are in the appendix.

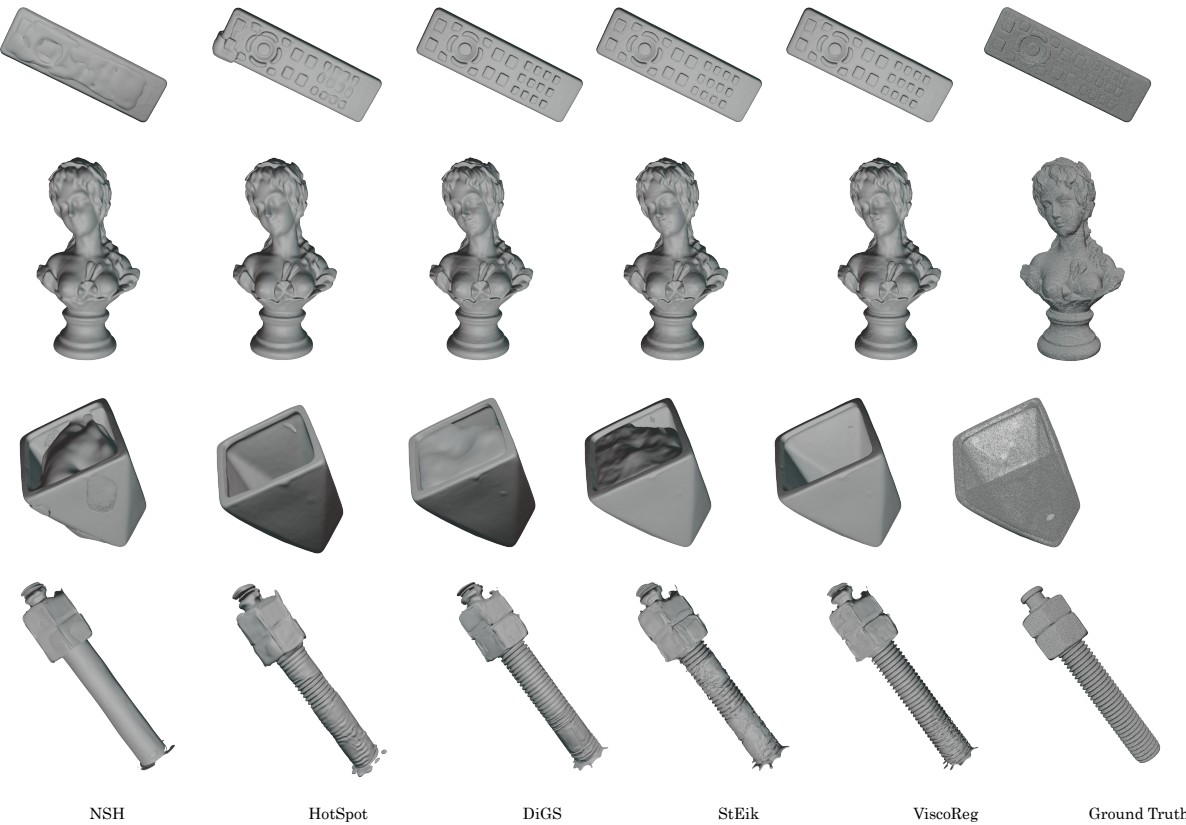

|  NSH | HotSpot | DiGS | StEik | ViscoReg | Ground Truth |

Figure 4: Results from real scans of Huang et al. (2024). ViscoReg recovers fine features such as buttons of the remote, threads of the screw, while also maintaining structural integrity of the empty flower pot. HotSpot and NSH struggle with details. DiGS and StEik recover lumpy artifacts.

## 6   Conclusion

We provide theoretical insight into improving the stability when learning an SDF using neural networks. We leverage classical PDE theory to provide an estimate on the worst-case error when using neural SDF methods. We also propose a physically-motivated regularizer (ViscoReg) for improved reconstruction. We demonstrate the effectiveness of our approach on many benchmarks and show improved performance compared to the state-of-the-art. However, we note that ViscoReg is designed to address training stability and high-frequency detail recovery in dense or noisy point clouds, rather than explicitly targeting completion of extremely sparse inputs or large missing regions. In regions far from the surface with low sample density (as in level-sets far from the surface in Fig. 1), approximation errors may persist due to limited network capacity and sample coverage. Empirically, ViscoReg's errors tend to appear in places far from boundary samples, and more broadly, thin structures remain a challenging setting for all neural SDF methods. While ViscoReg theoretically stabilizes the Eikonal gradient flow, escape from arbitrary local minima remains dependent on the choice of optimizer, initialization, and training schedule.

**Broader Impact Statement**

The research presented in this paper does not make use of human subjects, nor does it make any datasets available with personally identifiable information. The theoretical work is purely foundational and we do not foresee any direct ethical impacts from it. The empirical work concerns Neural SDF which aids in 3D surface reconstruction. Potential negative applications include the generation of deepfakes, and unauthorized reconstruction of private places. Despite these risks, we believe there is significant positive impact from this work in fields like robotics, content creation, and scientific visualization.

**Acknowledgments**

Our code is available at `https://github.com/mkrishnan9/ViscoReg`. This work was supported by ONR Award N00014-23-1-2086.

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

# A    Appendix

In this section, we provide supplementary details for our paper.

Let $\Omega \subseteq \mathbb{R}^n$ be an open, bounded domain and let $u : \Omega \to \mathbb{R}$ be a sufficiently regular function. The following norms are defined (Evans, 2022).

**Function Space Norms**

$L^p$ **Norm**    For $1 \le p < \infty$, the $L^p(\Omega)$ norm is defined as:

$$\|u\|_{L^p(\Omega)} = \left( \int_\Omega |u(x)|^p \, dx \right)^{1/p}.$$

For $p = \infty$, the $L^\infty(\Omega)$ norm is defined by the essential supremum:

$$\|u\|_{L^\infty(\Omega)} = \operatorname*{ess\,sup}_{x \in \Omega} |u(x)|.$$

$W^{k,p}$ **(Sobolev) Norm**    Let $k \in \mathbb{N}$ and $1 \le p \le \infty$. The Sobolev norm for the space $W^{k,p}(\Omega)$ is defined using multi-index notation for weak derivatives $D^\alpha u$, where $|\alpha| \le k$. For $1 \le p < \infty$, the norm is:

$$\|u\|_{W^{k,p}(\Omega)} = \left( \sum_{|\alpha| \le k} \|D^\alpha u\|_{L^p(\Omega)}^p \right)^{1/p}.$$

For $p = \infty$, the norm is:

$$\|u\|_{W^{k,\infty}(\Omega)} = \max_{|\alpha| \le k} \|D^\alpha u\|_{L^\infty(\Omega)}.$$

$C^k$ **Norm**    For a function $u \in C^k(\bar{\Omega})$, which is $k$ times continuously differentiable on the closure of $\Omega$, the $C^k$ norm is defined as the sum of the supremum norms of all its partial derivatives up to order $k$:

$$\|u\|_{C^k(\bar{\Omega})} = \sum_{|\alpha| \le k} \sup_{x \in \bar{\Omega}} |D^\alpha u(x)|.$$

## A.1    Viscosity Solutions

Denote by $\mathrm{USC}(\bar{\Omega})$ and $\mathrm{LSC}(\bar{\Omega})$, the space of upper and lower semi-continuous functions, respectively. The viscosity solution of the Eikonal equation is defined rigorously below.

**Definition A.1 (Viscosity Solution)** *A function $u \in \mathrm{USC}(\bar{\Omega})$ is a* viscosity subsolution *of (7) if for all $x_0 \in \bar{\Omega}$ and all $\phi \in C^\infty(\mathbb{R}^3)$ such that $u - \phi$ has a local maximum at $x_0$, we have:*

$$\begin{cases} \|\nabla\phi(x_0)\|_2 - f(x_0) \le 0, & \text{if } x_0 \in \Omega, \\ \min\{\|\nabla\phi(x_0)\|_2 - f(x_0), u(x_0) - g(x_0)\} \le 0, & \text{if } x_0 \in \partial\Omega. \end{cases}$$

*Similarly, $u \in \mathrm{LSC}(\bar{\Omega})$ is a* viscosity supersolution *of equation 7 if for all $x_0 \in \bar{\Omega}$ and all $\phi \in C^\infty(\mathbb{R}^3)$ such that $u - \phi$ has a local minimum at $x_0$, the following inequality holds:*

$$\begin{cases} \|\nabla\phi(x_0)\|_2 - f(x_0) \ge 0, & \text{if } x_0 \in \Omega, \\ \max\{\|\nabla\phi(x_0)\|_2 - f(x_0), u(x_0) - g(x_0)\} \ge 0, & \text{if } x_0 \in \partial\Omega. \end{cases}$$

*Then, $u \in C(\Omega)$ is a* viscosity solution *of (7) if it is both a viscosity subsolution and a supersolution.*

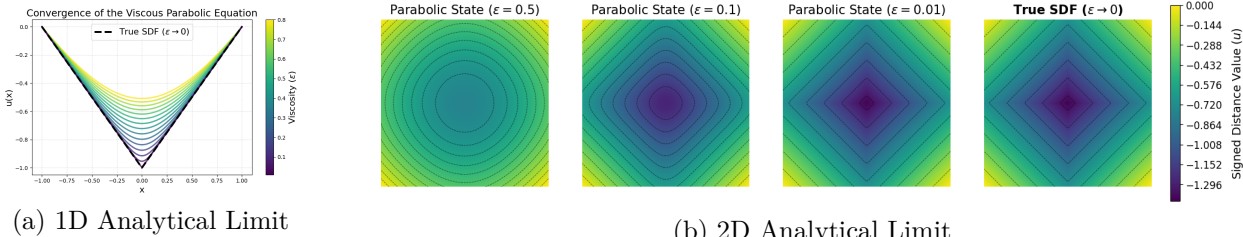

(a) 1D Analytical Limit

(b) 2D Analytical Limit

Figure 5: Exact analytical solutions demonstrating the vanishing viscosity limit. In 1D **(a)** and in 2D **(b)**, the intermediate parabolic solutions uniformly converge to the true SDF .

Next, we state formally the result for convergence of solutions of the parabolic equation 2 to the solution of equation 1 in the limit (Crandall & Lions, 1983).

**Theorem 3** *For each $\varepsilon > 0$, let $u_\varepsilon \in C^2(\bar{\Omega}) \cap C(\bar{\Omega})$ denote the unique solution to equation 2. Then $u_\varepsilon \to u$ uniformly, as $\varepsilon \to 0^+$, where $u$ is the unique viscosity solution of (1).*

To see this convergence, we plot intermediate parabolic solution and the viscosity solution for Hamilton Jacobi equations. For purely illustrative purposes, we visualize solutions of the closely related inviscid squared Eikonal $\|\nabla u\|^2 = 1$ and the corresponding viscous equation:

$$-\epsilon \Delta u + \|\nabla u\|^2 = 1, \tag{19}$$

which admits closed-form solutions via the Cole-Hopf transformation. Clearly, the squared Eikonal shares the same vanishing-viscosity limit (the true SDF) as the non-squared Eikonal equation ($-\epsilon \Delta u + \|\nabla u\| = 1$), but a closed-form solution via Cole–Hopf is available only for the viscous squared form. We therefore adopt this Hamilton Jacobi equation for the example.

For the 1D domain $x \in [-1, 1]$, the parabolic equation $-\epsilon u'' + (u')^2 = 1$ yields the closed-form solution:

$$u_\epsilon(x) = \epsilon \ln\left(\frac{\cosh(1/\epsilon)}{\cosh(x/\epsilon)}\right). \tag{20}$$

As demonstrated in Figure 5(a), for larger values of $\epsilon$, the solution forms a smooth, rounded curve. As $\epsilon \to 0$, this converges to the true SDF, $u(x) = 1 - |x|$. Also consider the 2D domain $(x, y) \in [-1, 1]^2$, where the solution to the parabolic equation is given by,

$$u_\epsilon(x, y) = \epsilon \ln\left(\frac{\cosh(c)}{\cosh(cx)}\right) + \epsilon \ln\left(\frac{\cosh(c)}{\cosh(cy)}\right). \tag{21}$$

with $c = (\epsilon\sqrt{2})^{-1}$. The corresponding level set plots in Figure 5(b) illustrate how the smoothed, dome-like geometries gradually sharpen. As $\epsilon \to 0$, we get the distance function, $u(x, y) = \sqrt{2} - \frac{|x| + |y|}{\sqrt{2}}$ and the level sets transition from circular to diamond-shaped. The zero-level set of the SDF is the diamond shape defined by $|x| + |y| = 2$.

## A.2 Mathematical Proofs

Let $u_1$, $u_2$ be viscosity solutions of $\|\nabla u\|_2 = f_1$, $\|\nabla u\|_2 = f_2$, respectively. The comparison principle states that if $f_1 \le f_2$ in $\bar{\Omega}$, and $u_{1|\partial\Omega} \le u_{2|\partial\Omega}$ then $u_1 \le u_2$ in $\bar{\Omega}$. We prove Lemmas 1 and 2 using this theory (Calder, 2018).

### A.2.1 Proof of Lemma 1

Let $C = \max_{\partial\Omega} \|u - v\|$. Then by definition:

$$u(x) - C \le v(x), \ x \in \partial\Omega. \tag{22}$$

The function $u(x) - C$ is also a solution to the equation $\|\nabla(u - C)\| = f$. The comparison principle for Hamilton-Jacobi equations (see Corollary 3.2 in Calder (2018)) then implies that:

$$u(x) - C \le v(x) \ x \in \Omega, \tag{23}$$

$$\implies u(x) - v(x) \le \max_{\partial\Omega}(u - v), \ \forall \ x \in \Omega. \tag{24}$$

This bound may also be obtained for $v - u$ by flipping $u$ and $v$. It follows that,

$$\|u - v\|_\infty \le \max_{\partial\Omega} \|u - v\| = \|g_1 - g_2\|_\infty. \tag{25}$$

### A.2.2 Proof of Lemma 2

Let $\hat{f}_1 = \lambda f_1$ where $\lambda = \max_\Omega \frac{f_2}{f_1}$. By construction, this ensures that $\hat{f}_1 \ge f_2$. Note that $\lambda u_1$ is the viscosity solution to the Eikonal equation with slowness $\hat{f}_1$. Since $\lambda u_1, u_2$ are the viscosity solutions, they obey the maximum principle, and hence $\lambda u_1 \ge u_2$. This leads to the following inequality:

$$u_2 - u_1 \le (\lambda - 1)u_1 \le \max_\Omega \frac{f_2 - f_1}{f_1} u_1$$

$$\le \frac{1}{C_f}\|f_1 - f_2\|_\infty u_1. \tag{26}$$

Since $u_1$ is the signed distance function, it can be bounded by the maximum time to travel between two points in the domain, and hence,

$$\|u_1\|_\infty \le C(\Omega)C_f^{-1}. \tag{27}$$

Inequality equation 26 may also be derived for $u_1 - u_2$ by swapping $u_1$ and $u_2$. Consequently:

$$\|u_1 - u_2\|_{L^\infty(\Omega)} \le C_\Omega C_f^{-2}\|f_1 - f_2\|_\infty. \tag{28}$$

### A.2.3 Proof of Theorem 1

First, we state the following classical result that follows from the Gagliardo–Nirenberg interpolation inequality relating different function norms Nirenberg (1959).

**Theorem 4** *Let $\Omega \subset \mathbb{R}^3$ be an open, smooth, bounded and connected domain. For $u \in L^1(\Omega) \cap W^{6,1}(\Omega)$, we have:*

$$\|u\|_\infty \le C_\Omega' \|u\|_{W^{6,1}(\Omega)}^{1/2} \|u\|_1^{1/2}. \tag{29}$$

*Here $C_\Omega'$ is a constant depending only on $\Omega$.*

Note that this result also holds for compact Riemannian manifolds Nirenberg (1959).

**Proof of Theorem 1** By Lemma 1, equation 1 can have at most one continuous viscosity solution. Since $u_{\theta^*}$ is smooth, it is the unique viscosity solution to equation 10. Define an auxiliary function $\hat{u}_{\theta^*} \in C(\bar{\Omega})$ such that it is the unique viscosity solution of the PDE:

$$\|\nabla \hat{u}_{\theta^*}(x)\|_2 = f_{\theta^*}, \ x \in \Omega, \quad \hat{u}_{\theta^*}(x) = 0, \ x \in \partial\Omega. \tag{30}$$

By the regularity of $\partial\Omega$ and $g_{\theta^*} : \partial\Omega \to \mathbb{R}$, we have $\hat{u}_{\theta^*} \in C(\Omega)$. Using the triangle inequality:

$$\|u - u_{\theta^*}\|_\infty \le \|u - \hat{u}_{\theta^*}\|_\infty + \|\hat{u}_{\theta^*} - u_{\theta^*}\|_\infty. \tag{31}$$

Using Lemma 2 and Lemma 1 to bound the first and second term, respectively:

$$\|u - u_{\theta^*}\|_\infty \le \|g_{\theta^*}\|_\infty + C_\Omega C_{\theta^*}^{-2}\|1 - f_{\theta^*}\|_\infty. \tag{32}$$

where $C_\Omega$ is a constant depending only on the domain. Using the Gagliardo-Nirenberg interpolation inequality (see Theorem 4 in the appendix) for the open bounded set $\Omega$ and compact Riemannian manifold $\partial\Omega$, along with Assumption 3:

$$\|u - u_{\theta*}\|_\infty \lesssim M_{\theta*}\|g_{\theta*}\|_1^{\frac{1}{2}} + M_{\theta*}C_{\theta*}^{-2}\|1 - f_{\theta*}\|_1^{\frac{1}{2}}, \tag{33}$$

where the hidden constants in $\lesssim$ only depends on $\bar\Omega$. Observe that both $\|g_{\theta*}\|_1$ and $\|1 - f_{\theta*}\|_1$ can be approximated by their sample norms. The neural SDF method samples uniformly in the domain for the Eikonal loss $\|1 - f_{\theta*}\|_1$ and hence the $L^1(\Omega)$ quadrature error is $\mathcal{O}(N^{-1/3})$, where $N$ is the number of sample points. Assumption 2 can be used to bound the boundary loss $\|g_{\theta*}\|_1$. This gives:

$$\|u - u_{\theta*}\|_\infty \lesssim M_{\theta*}\left(\frac{\sum_{i=1}^N |g_{\theta*}(x_i)|}{N}\right)^{\frac{1}{2}} + M_{\theta*}C_{\theta*}^{-2}\left(\frac{\sum_{i=1}^{M+N} |1 - f_{\theta*}|}{M + N}\right)^{\frac{1}{2}}$$
$$+ \mathcal{O}(N^{-\frac{\beta}{2}}) + \mathcal{O}((M + N)^{-1/6}). \tag{34}$$

Since the first term can be represented using the boundary loss, and the second term by the Eikonal loss, we obtain the required result. $\square$

The $L^2$ error estimate may be obtained in a similar setting as Theorem 2 by using a more general version of Theorem 1 that we state below.

**Theorem 5** *Nirenberg (1959) Let $\Omega \subset \mathbb{R}^3$ be an open smooth connected domain. Let $1 \le r, m \le \infty$ and $\alpha \in [0, 1]$ such that:*

$$(1 - \alpha)\left(\frac{m}{3} - \frac{1}{r}\right) = \frac{\alpha}{p}, \tag{35}$$

*for $p = 1, 2$. Then for $u \in L^2(\Omega) \cap W^{m,r}(\Omega)$, we have:*

$$\|u\|_\infty \le \|u\|_{W^{m,r}(\Omega)}^{1-\alpha} \|u\|_p^\alpha. \tag{36}$$

By following the proof of Theorem 2, with this inequality, we can provide a similar result.

### A.2.4 Discussion on Assumptions

**Empirical verification of Assumption 1.** The ground truth SDF satisfies the Eikonal equation ($\|\nabla u\|_2 = 1$ almost everywhere). Assumption 1 posits that the network optimization has converged to a sufficiently good local minimum where its gradients approximate that of the target function by being bounded away from zero. We conducted an experiment to confirm that our trained networks do, in fact, achieve these non-zero gradients in practice. We evaluate the gradients on a dense uniform grid of 512x512 points for the fractal shape in Fig. 1. The median of the gradient norms is 0.9950, with no points having zero gradient norm and it is bounded uniformly away from zero.

Table 5: Gradient norm statistics for the converged ViscoReg network, evaluated on a dense uniform grid of $512^2$ points for the Mandelbrot set reconstruction.

| # Points | Mean | Median | Std | Min | $\% < 0.01$ | $\% = 0.0$ |
|---|---|---|---|---|---|---|
| 262,144 | 0.9739 | 0.9950 | 0.1356 | 0.0146 | 0.0% | 0.0% |

**Assumption 2.** Assumption 2 describes the sampling density of the input point cloud, which is a property of the dataset's resolution. It is not a property of the network or the training dynamics. Generalization error estimates are derived for the continuous functions, whereas the neural network is trained on the loss function evaluated at discrete sample points. To relate the two, we must establish a relationship between the discrete sampling sum and the continuous integral, and Assumption 2 provides this bridge.

To examine the dependence of reconstruction quality on the number of manifold points, we evaluate ViscoReg on SRB with the training point cloud subsampled to 10%, 40%, 60%, and 80% of the original. Results are reported in Table 6. As expected, reconstruction quality degrades gracefully with fewer points. We note that even at 40% retention, ViscoReg remains competitive with several baselines trained on the full point cloud.

Table 6: Ablation on manifold point density for SRB. $d_C$: Chamfer, $d_H$: Hausdorff distance. Even when trained with 40% manifold points, ViscoReg remains competitive with several baselines trained on the full point cloud (cf. Table 1).

| Manifold Points | $d_C \downarrow$ | $d_H \downarrow$ |
|---|---|---|
| 10% ($\sim$5k–9k pts) | 0.292 | 4.37 |
| 40% | 0.227 | 4.36 |
| 60% | 0.198 | 3.71 |
| 80% | 0.194 | 3.66 |
| 100% (Baseline) | **0.176** | **2.76** |

A thorough investigation of optimal sampling strategies and convergence rates for Neural SDFs is an important direction that merits dedicated study; we refer the interested reader to Lin et al. (2024); Pais et al. (2024) for recent work on this topic.

### A.3 Derivation of the Gradient Flow

For $p = 1$, the first variation of the energy $\delta\mathcal{L}_{veik}(u)[v]$ in the direction of a test function $v$ is:

$$\delta\mathcal{L}_{veik}(u)[v] = \int_\Omega \text{sign}(r(u)) \left( \frac{\nabla u \cdot \nabla v}{\|\nabla u\|_2} - \varepsilon\Delta v \right) dx. \tag{37}$$

To obtain the Fréchet derivative, we apply integration by parts (with vanishing boundary terms as SDF is zero on the boundary). Let $r(u) = \|\nabla u\|_2 - 1 - \varepsilon\Delta u$ denote the viscous residual.

- Eikonal term: $\int \text{sign}(r)\frac{\nabla u \cdot \nabla v}{\|\nabla u\|} dx = -\int v\nabla \cdot \left(\text{sign}(r)\frac{\nabla u}{\|\nabla u\|}\right) dx.$

- Viscous term: $-\varepsilon \int \text{sign}(r)\Delta v \, dx = -\varepsilon \int v\Delta(\text{sign}(r))dx.$

The Fréchet derivative is:

$$\frac{\delta\mathcal{L}_{veik}}{\delta u} = -\nabla \cdot \left( \text{sign}(r)\frac{\nabla u}{\|\nabla u\|_2} \right) - \varepsilon\Delta(\text{sign}(r)), \tag{38}$$

and the gradient flow equation may be obtained as $u_t = -\frac{\delta\mathcal{L}}{\delta u}$.

### A.4 Proof of stability for $p = 2$.

For $p = 2$, the first variation of the energy $\delta\mathcal{L}_{veik}(u)[v]$ is:

$$\delta\mathcal{L}_{veik}(u)[v] = \int_\Omega r(u) \left( \frac{\nabla u \cdot \nabla v}{\|\nabla u\|_2} - \varepsilon\Delta v \right) dx. \tag{39}$$

Applying integration by parts as before

- Eikonal term: $\int r\frac{\nabla u \cdot \nabla v}{\|\nabla u\|} dx = -\int v\nabla \cdot \left(r\frac{\nabla u}{\|\nabla u\|}\right) dx.$

- Viscosity term: $-\varepsilon \int r\Delta v \, dx = -\varepsilon \int v\Delta r \, dx.$

Table 7: Quantitative comparison of SDF reconstruction on 2D Mandelbrot Set.

| | Overall | | Near Surface ($< 0.05$) | |
|---|---|---|---|---|
| Method | RMSE | MAE | RMSE | MAE |
| SIREN | 0.055 | 0.036 | 0.0150 | 0.0091 |
| DiGS | 0.052 | 0.031 | 0.0110 | 0.0074 |
| StEik | 0.034 | 0.023 | 0.0120 | 0.0076 |
| ViscoReg | **0.024** | **0.015** | **0.0091** | **0.0062** |

The Fréchet derivative is:

$$\frac{\delta \mathcal{L}_{veik}}{\delta u} = -\nabla \cdot \left( r \frac{\nabla u}{\|\nabla u\|_2} \right) - \varepsilon \Delta r. \tag{40}$$

The gradient flow equation is then $u_t = -\frac{\delta \mathcal{L}}{\delta u}$. Linearizing around $u_0 = \mathbf{a} \cdot x$ (with $\|\mathbf{a}\|_2 = 1$):

$$u_t = (\mathbf{a} \cdot \nabla)^2 u - \varepsilon^2 \Delta^2 u. \tag{41}$$

In the Fourier domain:

$$\hat{v}_t = - \left( (\mathbf{a} \cdot \omega)^2 + \varepsilon^2 \|\omega\|_2^4 \right) \hat{u}. \tag{42}$$

The Fourier symbol $\lambda(\omega) = -[(\mathbf{a} \cdot \omega)^2 + \varepsilon^2 \|\omega\|_2^4]$ is always non-positive. Frequency damping and thus stability is again guaranteed by the $\varepsilon^2 \|\omega\|^4$ term.

### A.5 Model Studies and Implementation Details

All the methods are evaluated on a single Nvidia RTX A6000 GPU. For testing for all shapes, we use the Marching Cubes algorithm Lorensen & Cline (1998) with resolution 512 and the same mesh extraction process as Yang et al. (2023), Ben-Shabat et al. (2022) and other methods.

#### A.5.1 2D Mandelbrot Set

All methods were evaluated with the same 4 layer 128 neuron architecture, and optimized for 10000 iterations. We compute the ground truth SDF using a dense sampling of the 2D domain and report the RMSE and MAE of the learned SDF.

All methods utilized a 4-layer SIREN architecture (sine activations) with 128 hidden units, initialized via Multi-Frequency Geometric Initialization (MFGI). The models were optimized using a learning rate of $\mathbf{5 \times 10^{-5}}$. At each step, we sampled $\mathbf{15,000}$ domain points. The loss terms were balanced with weights $[\alpha_m, \alpha_{nm}, \alpha_e] = [3000, 100, 50]$. To decouple spurious errors from far-field extrapolation, we report RMSE and MAE within a bounded domain (all points with SDF $<0.5$). As the object is normalized to have a bounding box half-width of 0.5, this evaluation threshold of $\delta = 0.5$ is equivalent to 100% of the object's maximal spatial extent from the center and includes all interior points. We additionally report "Near Surface" error (i.e abs(SDF)$<0.05$) to highlight accuracy near the zero-level set. See Table 7 for results.

#### A.5.2 Surface Reconstruction Benchmark

First, we center the input point clouds at the origin and normalize them so that it is inside the unit cube. The bounding box is scaled to 1.1 times the size of the shape. At each iteration, we sample 15,000 points from the original point cloud and an additional 15,000 points uniformly from the bounding box. Training is conducted for 10,000 iterations with a learning rate of $10^{-4}$. The weights were taken to be $[\alpha_m, \alpha_{nm}, \alpha_e] = [3000, 100, 50]$. Baseline decay for all shapes is initial $\varepsilon = 0.5$, decayed linearly at 20/40/60/80% iterations to 0.4/0.04/0.005/0. We used 5 hidden layers, and 128 nodes. MFGI with sphere initial parameters was taken to be $(1.6, 0.1)$.

Additional quantitative results for each individual shape are presented in Table 17.

**Adaptive Decay Strategy:** We test a method to adaptively control the $\varepsilon$ parameter based on the absolute error of the eikonal constraint. The update for the viscosity weight $\varepsilon_t$ at iteration $t$ is defined as:

$$\varepsilon_t = \begin{cases} \max\left(\varepsilon_{\min}, \varepsilon_{\text{target\_decay}}(t) + S \cdot \text{EMA}_{t-1}\right) & \text{if } t < 0.5 \cdot N, \\ 0 & \text{if } t \geq 0.5 \cdot N. \end{cases}$$

Here $\varepsilon_{\text{target\_decay}}$ is the baseline target decay. The baseline provides a monotonically decaying target for $\varepsilon_t$. The progress $p$ is normalized over the first 50% of training: $p = t/(0.5 \cdot N)$, with $\varepsilon_{\text{target\_decay}}(t) = \varepsilon_{\text{initial}} \cdot (\gamma_{\text{factor}})^p$. Then we use Exponential Moving Average (EMA) to track the recent residual of the pure Eikonal constraint, $L_{\text{Eik}}$, to assess the need for regularization.

$$\text{EMA}_t = \beta \cdot \text{EMA}_{t-1} + (1 - \beta) \cdot L_{\text{Eik},t}.$$

For the SRB dataset we use $\varepsilon_{\text{initial}} = 0.4$, $\gamma_{\text{factor}} = 0.01$, $\beta = 0.999$, $S = 0.5$ (Sensitivity factor) and $\varepsilon_{\min} = 0.0$.

Table 8: Adaptive Decay for SRB. $d_C$ : Chamfer and $d_H$: Hausdorff distance.

| Shape | Piecewise Linear | | Adaptive Decay | |
|---|---|---|---|---|
| | $d_C$ | $d_H$ | $d_C$ | $d_H$ |
| anchor | 0.23 | 4.35 | 0.23 | 4.14 |
| dc | 0.16 | 1.33 | 0.15 | 1.53 |
| daratech | 0.18 | 2.96 | 0.19 | 1.87 |
| gargoyle | 0.18 | 3.81 | 0.18 | 4.31 |
| lord_quas | 0.13 | 1.37 | 0.13 | 1.13 |
| **Mean** | **0.18** | **2.76** | **0.18** | **2.60** |

### A.5.3 $L^1$ vs $L^2$ ViscoReg

We tested the model on SRB using $p = 2$ for the ViscoReg parameter in Table 9. Empirically, we see better performance for $p = 1$. The improved performance of $p = 1$ is seen across a range of method such as DiGS and StEik as well. It is perhaps unsurprising as $L^2$ tends to suppress the large gradients required for sharp edges, which can cause over-smoothing. Whereas the $L^1$ norm is more tolerant to these geometric features.

### A.5.4 Computational Cost

The bulk of the computational cost comes from calculating the Laplacian. Hence, the computational cost is just slightly lower than that of DiGS and higher than that of StEik (who computes the directional derivative with more efficient calculations). Timings are presented in Table 10 for the SRB dataset with experiments conducted on A100 GPU.

### A.5.5 Ablation

For the ablation studies, the decay schedules are as follows. BL$\times x$= baseline decay of 0.5/0.4/0.04/0.005/0 at 0/20/40/60/80% iterations scaled by $x$. Fast decay corresponds to a quick decay to 0, of 0.5/0.0 at 0/20% iterations. Slow decay corresponds to extended decay at 90 percent of iterations with a schedule 0.5/0.4/0.04/0.005/0 at 0/20/40/60/90%. We also test piecewise constant and piecewise quintic decay as opposed to piecewise linear. Ablation studies per shape are provided in Tab.11-15.

| Loss | $d_C$ | $d_H$ |
|---|---|---|
| $L^1$ ViscoReg | 0.18 | 2.76 |
| $L^2$ ViscoReg | 0.19 | 4.49 |

Table 9: $L^1$ vs $L^2$ ViscoReg loss terms

Table 10: Runtime comparison in ms. All models utilize a $4 \times 256$ architecture with 0.26M parameters.

| Method | HotSpot | DiGS | StEik | ViscoReg |
|---|---|---|---|---|
| **Runtime (ms)** | 20.43 | 36.05 | 27.68 | 35.50 |

Table 11: Ablation on $\varepsilon$ decay for `anchor`.

| Method | $d_C \downarrow$ | $d_H \downarrow$ |
|---|---|---|
| BL | **0.23** | 4.35 |
| BL $\times 2$ | **0.25** | 5.36 |
| BL $\times 0.5$ | 0.26 | 4.70 |
| Piecewise Const. | 0.29 | 7.97 |
| Quintic | 0.26 | 6.35 |
| Fast decay (0 @ 20%) | 0.31 | 7.33 |
| Slow decay(0 @ 90%) | 0.25 | 5.71 |
| $\varepsilon = 0$ (SIREN [47]) | 0.72 | 10.98 |
| SoTA best | 0.26 | **4.26** |

Table 12: Ablation on $\varepsilon$ decay for `dc`.

| Method | $d_C \downarrow$ | $d_H \downarrow$ |
|---|---|---|
| BL | 0.16 | 1.33 |
| BL $\times 2$ | **0.15** | 1.44 |
| BL $\times 0.5$ | 0.16 | 1.39 |
| Piecewise Const. | 0.16 | 1.49 |
| Quintic | 0.17 | 1.32 |
| Fast decay (0 @ 20%) | 0.16 | 1.35 |
| Slow decay(0 @ 90%) | 0.15 | **1.23** |
| $\varepsilon = 0$ (SIREN [47]) | 0.34 | 6.27 |
| SoTA best | 0.15 | 1.70 |

Table 13: Ablation on $\varepsilon$ decay for `daratech`.

| Method | $d_C \downarrow$ | $d_H \downarrow$ |
|---|---|---|
| BL | **0.18** | 1.33 |
| BL $\times 2$ | **0.18** | 1.44 |
| BL $\times 0.5$ | 0.20 | 1.39 |
| Piecewise Const. | 0.21 | 1.49 |
| Quintic | 0.19 | 1.32 |
| Fast decay (0 @ 20%) | 0.19 | 1.35 |
| Slow decay(0 @ 90%) | 0.20 | **1.23** |
| $\varepsilon = 0$ (SIREN [47]) | 0.21 | 6.27 |
| SoTA best | **0.18** | 1.72 |

Table 14: Ablation on $\varepsilon$ decay for `gargoyle`.

| Method | $d_C \downarrow$ | $d_H \downarrow$ |
|---|---|---|
| BL | 0.18 | **3.81** |
| BL $\times 2$ | **0.17** | 3.97 |
| BL $\times 0.5$ | 0.21 | 9.18 |
| Piecewise Const. | 0.18 | 4.09 |
| Quintic | 0.19 | 6.06 |
| Fast decay (0 @ 20%) | 0.18 | 3.95 |
| Slow decay(0 @ 90%) | 0.19 | 4.48 |
| $\varepsilon = 0$ (SIREN [47]) | 0.46 | 7.76 |
| SoTA best | **0.17** | 4.10 |

Table 15: Ablation on $\varepsilon$ decay for `lord_quas`.

| Method | $d_C \downarrow$ | $d_H \downarrow$ |
|---|---|---|
| BL | 0.13 | 1.37 |
| BL $\times 2$ | 0.13 | 2.18 |
| BL $\times 0.5$ | 0.14 | 6.45 |
| Piecewise Const. | 0.14 | 3.65 |
| Quintic | 0.14 | 2.30 |
| Fast decay (0 @ 20%) | 0.13 | 2.04 |
| Slow decay(0 @ 90%) | 0.12 | **1.41** |
| $\varepsilon = 0$ (SIREN [47]) | 0.35 | 8.96 |
| SoTA best | **0.11** | **0.70** |

Further ablation studies are provided for real nonsynthetic data in Table 16. Baseline decay is 1.0 decayed linearly at 60% of iterations to 0. BL$\times x$ denotes baseline scaled by $x$. Lower start starts decay at 0.8 and higher start starts decay at 1.2.

|  | $d_C \times 10^{-2}$ | F-Score | NCS |
|---|---|---|---|
| Baseline | 30.55 | 86.50 | 95.67 |
| Faster Decay (0 @50) | 33.88 | 84.44 | 92.61 |
| Slower Decay (0 @70) | 36.88 | 85.07 | 93.36 |
| Lower Start at 0.8 | 35.42 | 85.10 | 94.08 |
| Higher start at 1.2 | 36.48 | 85.00 | 91.64 |
| BL×0.5 | 35.58 | 84.42 | 91.82 |

Table 16: Ablation on real data from Huang et al. (2024).

| Shape | Method | $d_C$ | $d_H$ | Shape | Method | $d_C$ | $d_H$ |
|---|---|---|---|---|---|---|---|
| Overall | IGR wo n | 1.38 | 16.33 | DC | IGR wo n | 0.63 | 10.35 |
|  | SIREN wo n | 0.42 | 7.67 |  | SIREN wo n | 0.34 | 6.27 |
|  | SAL | 0.36 | 7.47 |  | SAL | 0.18 | 3.06 |
|  | IGR+FF | 0.96 | 11.06 |  | IGR+FF | 0.86 | 10.32 |
|  | PHASE+FF | 0.22 | 4.96 |  | PHASE+FF | 0.19 | 4.65 |
|  | DiGS | 0.19 | 3.52 |  | DiGS | **0.15** | 1.70 |
|  | StEik | **0.18** | 2.80 |  | StEik | 0.16 | 1.73 |
|  | ViscoReg | **0.18** | 2.76 |  | ViscoReg | 0.16 | 1.33 |
|  | ViscoReg (quad) | **0.18** | **2.69** |  | ViscoReg (quad) | 0.16 | **1.29** |
| Anchor | IGR wo n | 0.45 | 7.45 | Gargoyle | IGR wo n | 0.77 | 17.46 |
|  | SIREN wo n | 0.72 | 10.98 |  | SIREN wo n | 0.46 | 7.76 |
|  | SAL | 0.42 | 7.21 |  | SAL | 0.45 | 9.74 |
|  | IGR+FF | 0.72 | 9.48 |  | IGR+FF | 0.26 | 5.24 |
|  | PHASE+FF | 0.29 | 7.43 |  | PHASE+FF | **0.17** | 4.79 |
|  | DiGS | 0.29 | 7.19 |  | DiGS | **0.17** | 4.10 |
|  | StEik | 0.26 | 4.26 |  | StEik | 0.18 | 4.49 |
|  | ViscoReg | **0.23** | **4.35** |  | ViscoReg | 0.18 | **3.80** |
|  | ViscoReg (quad) | 0.26 | 4.90 |  | ViscoReg (quad) | 0.18 | 4.15 |
| Daratech | IGR wo n | 4.9 | 42.15 | Lord Quas | IGR wo n | 0.16 | 4.22 |
|  | SIREN wo n | 0.21 | 4.37 |  | SIREN wo n | 0.35 | 8.96 |
|  | SAL | 0.62 | 13.21 |  | SAL | 0.13 | 4.14 |
|  | IGR+FF | 2.48 | 19.6 |  | IGR+FF | 0.49 | 10.71 |
|  | PHASE+FF | 0.35 | 7.24 |  | PHASE+FF | **0.11** | **0.71** |
|  | DiGS | 0.20 | 3.72 |  | DiGS | 0.12 | 0.91 |
|  | StEik | 0.18 | 1.72 |  | StEik | 0.13 | 1.81 |
|  | ViscoReg | 0.19 | 2.97 |  | ViscoReg | 0.14 | 1.37 |
|  | ViscoReg (quad) | **0.17** | **1.43** |  | ViscoReg (quad) | 0.13 | 1.69 |

Table 17: Additional quantitative results on the Surface Reconstruction Benchmark using point data without normals.

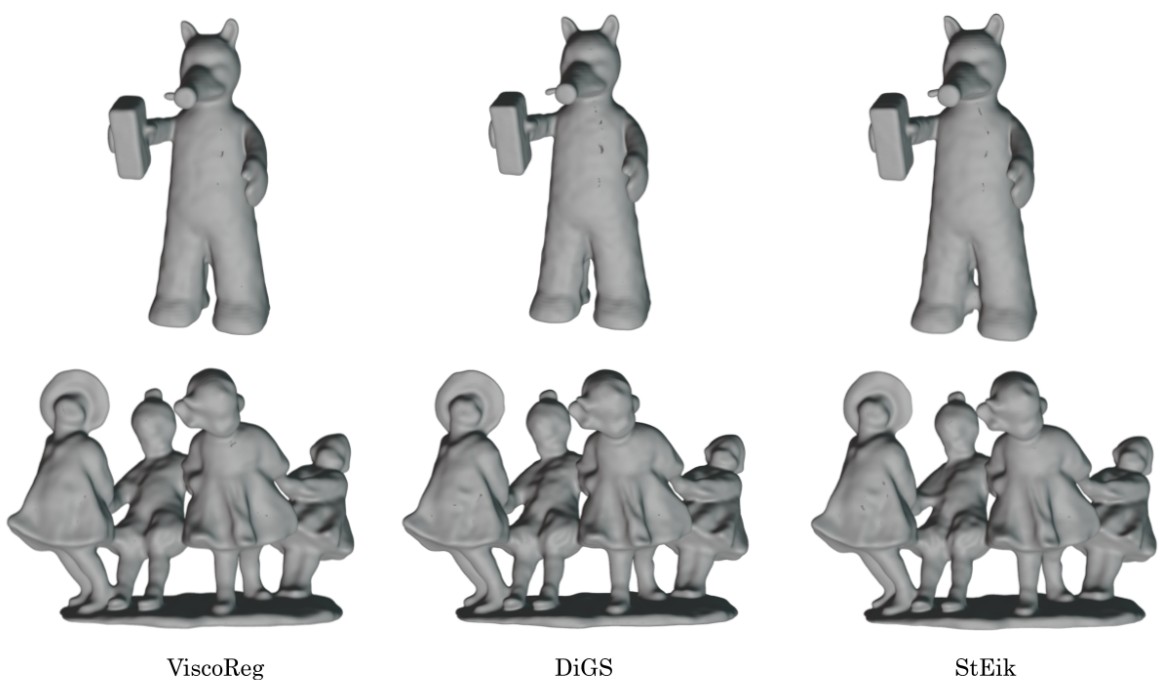

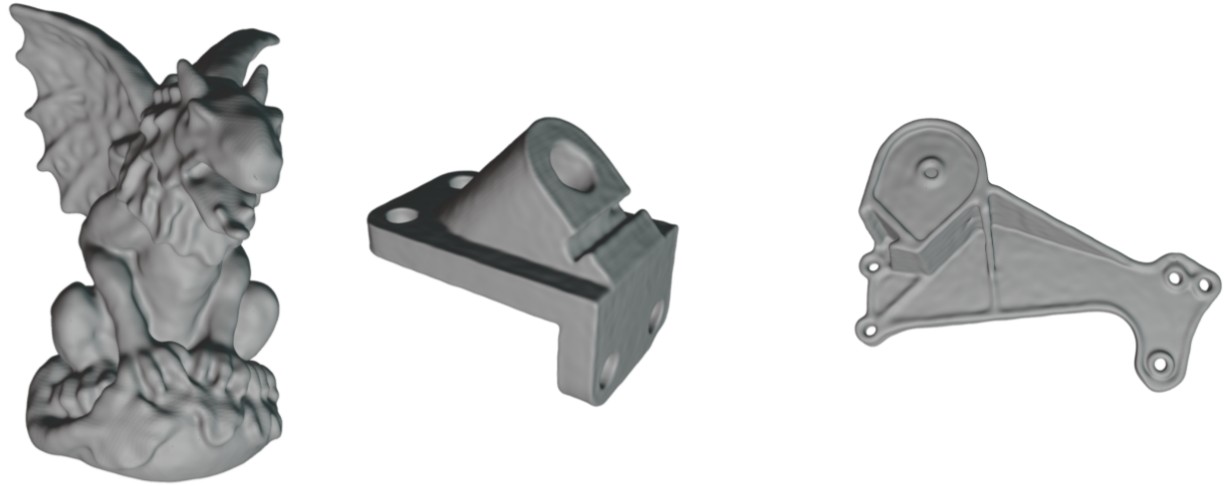

(a) Comparison on SRB shapes `dc` and `lord_quas`

(b) Reconstructed shapes `gargoyle`, `anchor`, and `daratech` using ViscoReg.

Figure 6: Qualitative results from SRB.

### A.5.6 Faster Convergence

We demonstrate ViscoReg's faster convergence to better minima (in terms of the Eikonal constraint) than SIREN Sitzmann et al. (2019b) with unstable Eikonal loss (see Fig 7).

### A.5.7 Scene Reconstruction

For this experiment, we used an architecture of 8 hidden layers, and 512 channels. At each iteration, we sample 15,000 points from the original point cloud and another 15,000 points uniformly at random within

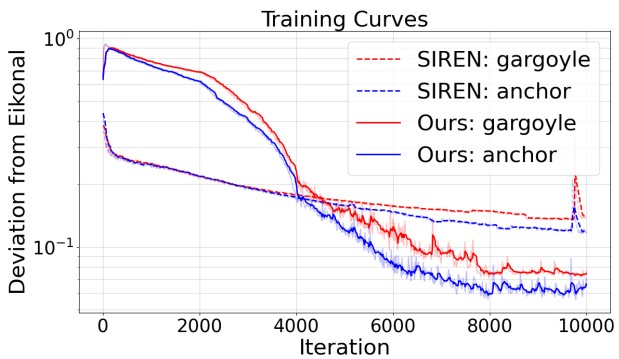

Figure 7: Deviation from Eikonal.

the bounding box. Training is performed for 100,000 iterations with a learning rate of $8 \times 10^{-6}$. The weights used were $[\alpha_m, \alpha_{nm}, \alpha_e] = [5000, 100, 50]$. The viscosity coefficient $\varepsilon$ decayed piecewise linearly starting at 0.5, decaying to 0.01 at 50 percent iterations followed by steeply decaying to 0 at 60 percent.

### A.5.8   Shapenet

We follow the preprocessing and evaluation methodology outlined in Williams et al. (2021). First, the preprocessing technique from Mescheder et al. (2019) is applied, then performance is evaluated on the first 20 shapes of the test set for each shape class. The preprocessing step extracts ground truth surface points from ShapeNet and generates random samples within the domain, and their corresponding occupancy values. We use the MFGI initialization proposed in DiGS for this experiment. For evaluation, the ground truth surface points are used to compute the squared Chamfer distance, while the labeled random samples are used to calculate the Intersection over Union (IoU).

During training, 15,000 points are sampled from the original point cloud and an additional 15,000 points are sampled uniformly at random within the bounding box. The model is trained for 10,000 iterations with a learning rate of $5 \times 10^{-5}$. The weights were chosen to be $[\alpha_m, \alpha_{nm}, \alpha_e] = [3000, 100, 50]$. The viscosity coefficient $\varepsilon$ decayed piecewise linearly starting at 1.0 decreasing at 10%, 20%, 30% and 40% to 0.0 for all shapes besides rifle, lamp, and table. For these shapes, we start the decay at $\varepsilon = 10.0$.

Note that to report results for HotSpot Wang et al. (2025), we used as reported in their work, 5 layer, 128 hidden dimension architecture with linear layers.

For quadratic ViscoReg architecture, the decay rate was taken as 1.0/0.5/0/0 at 0/10/20% iterations.

Additional qualitative results are provided in Figure 9.

### A.5.9   Real scans

To test the performance of this method on a dataset of real noisy scans, we used a 4 layer, 256 hidden dimension architecture for all methods. All methods were trained for 20000 iterations. We used standard siren initialization for all methods with a learning rate of 1e-4 for ViscoReg, StEik, DiGS and SIREN and learning rate 5e-5 for all others (based on what yielded the best performance). A single RTX2080Ti GPU was used for all experiments. Additional result on the `marker` shape is presented in Fig. 10 where all methods with the exception of ViscoReg struggle to maintain the hollow interior of the mug.

### A.5.10   Activation Function Ablation

We use SIREN as the backbone architecture. We utilize this architecture because our method relies on second-order derivatives (Laplacians), and ReLU networks have vanishing second-order gradients being piecewise linear. SIREN remains the standard architecture for many recent SoTA; for instance, HotSpot, NeurCADRecon, StEik, etc. all rely on it as their main architecture.

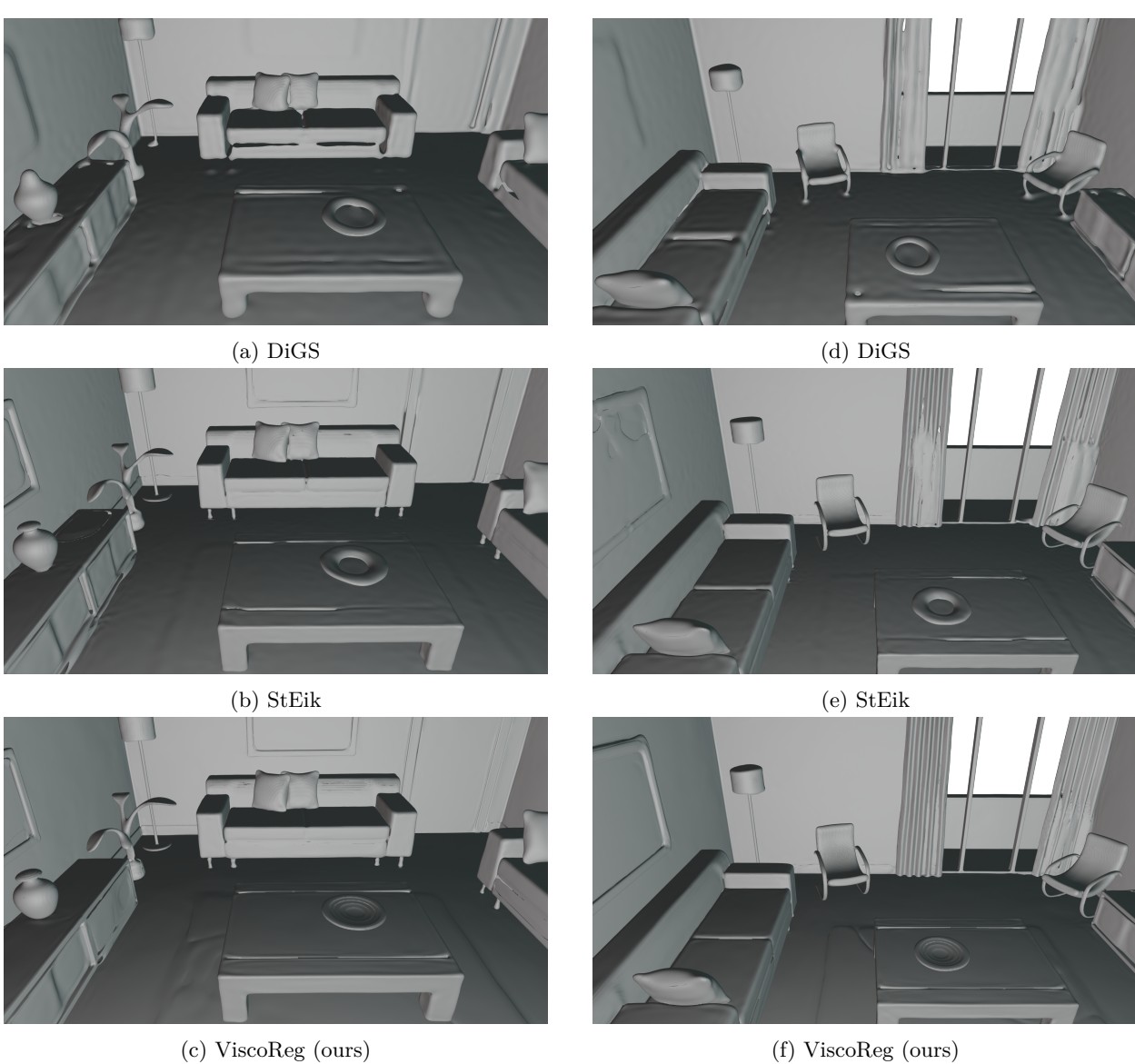

Figure 8: Results from the scene reconstruction benchmark from Sitzmann et al. (2019b). The left column compares results on one view of the scene: The DiGS mesh (a) is missing fine details like the sofa legs and picture frame details. StEik (b) performs better but struggles with fine details such as the curtains. ViscoReg (c) reconstructs these fine details with high fidelity. The right column provides additional views of the scene.

However, as proof of concept that ViscoReg can be applied to other activation functions (provided they support second-order derivatives), we evaluated ViscoReg using the FINER activation (Liu et al., 2024) on the SRB dataset. We kept the architecture and hyperparameters consistent with the SIREN experiments.

Table 18: Comparison of StEik and ViscoReg models

|  | SIREN | | FINER | |
| --- | --- | --- | --- | --- |
| Model | $d_C$ | $d_H$ | $d_C$ | $d_H$ |
| StEik (lin) | 0.20 | 4.56 | 0.28 | 9.59 |
| ViscoReg (Ours) | **0.18** | **2.76** | **0.26** | **5.55** |

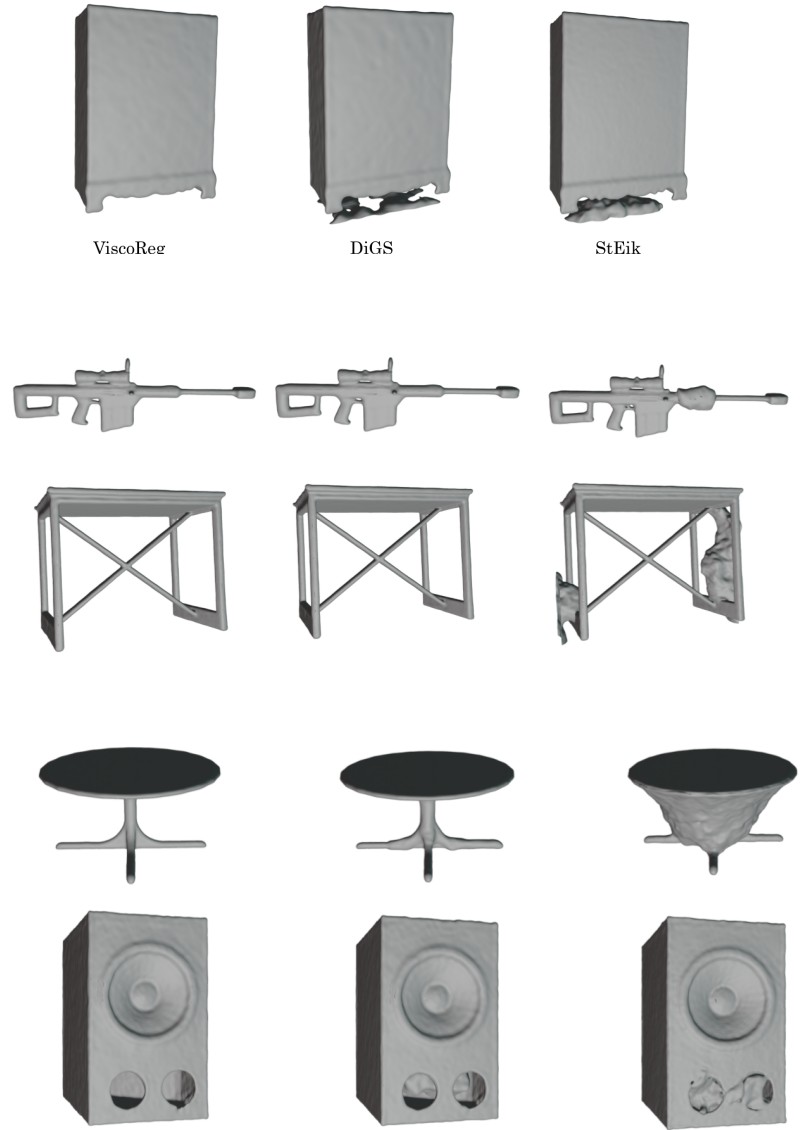

Figure 9: Qualitative results from the ShapeNet dataset from bench, cabinet, rifle and table categories. Chang et al. (2015).

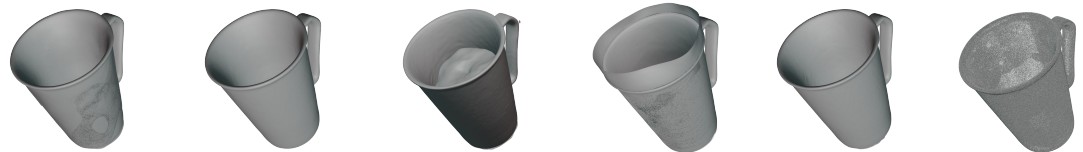

Figure 10: Further Results from real scan reconstruction.

### A.5.11 Figure 1

We include zoomed-in views of Figure 1 for a closer inspection.

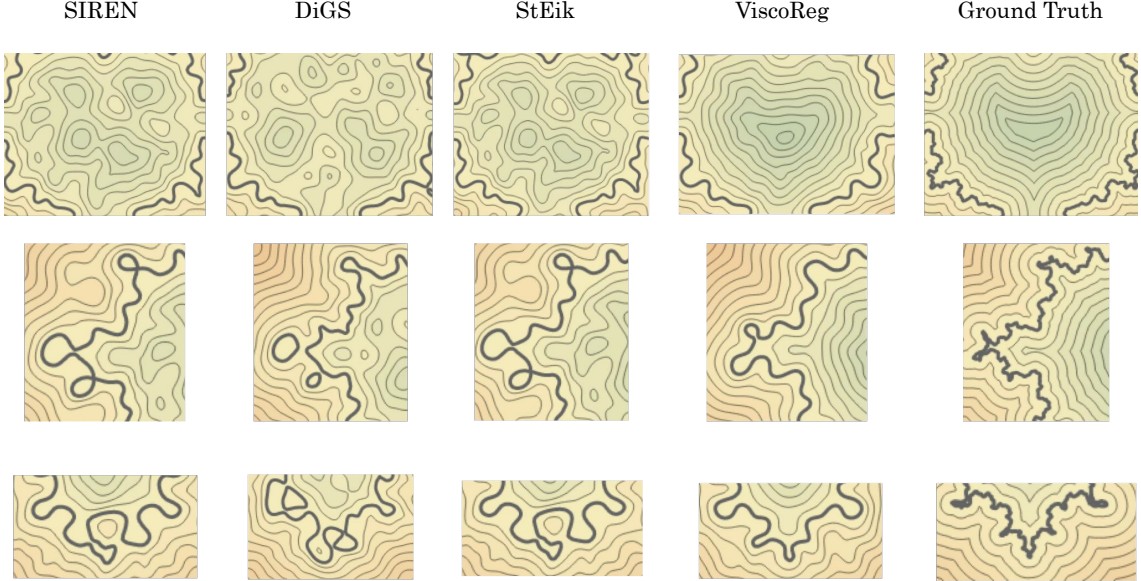

| SIREN | DiGS | StEik | ViscoReg | Ground Truth |

Figure 11: Zoomed-in views, where ViscoReg outperforms baselines.

### A.5.12 1D Sanity Check for Neural SDF

We train a 46 parameter SIREN model to approximate the Signed Distance Function over the domain $[-0.5, 0.5]$. The network correctly recovers the function, and the convergence of the Eikonal and manifold loss at the two boundary points are also plotted in Figure 12.

To empirically test whether the Neural SDF framework can escape suboptimal configurations, we pre-train a small SIREN network for 10,000 iterations to fit piecewise-linear zig-zag functions (a W-shape and a M-shaped zig-zag function) via exact pointwise MSE supervision over a dense grid. We then train the network for 30,000 iterations. As shown in Figure 13, the network escapes the zig-zag initialization and converges to the correct V-shaped SDF in both cases. During the initial iterations after switching to Neural SDF training, the loss temporarily increases as the network departs from the pre-trained zig-zag configuration. The function progressively smooths before converging to the correct V-shaped SDF. We note that the loss temporarily increases during the first few iterations as the network departs from the pre-trained zig-zag configuration. This is also observed across other datasets (see Fig. 7). This is reasonable in nonconvex optimization with competing loss terms, where reducing one term may temporarily increase another.

The escape trajectory observed in Figure 13 also depends on the annealing schedule of the viscosity coefficient $\epsilon$. Initially, when $\epsilon$ is relatively large, the parabolic diffusion term $\epsilon \Delta u$ helps diffuse the kinks of the zig-zag initialization and the function becomes smoother. As a consequence, the inviscid Eikonal loss temporarily increases during the early iterations (as seen in the loss curves) because the network is being actively pulled out of its suboptimal Lipschitz configuration into a parabolic state. As $\epsilon$ decays over the annealing phase, the influence of diffusion wanes, and the Eikonal term drives the now-smoothed function to sharpen into the correct V-shaped target SDF.

This 1D experiment is provided as an empirical illustration of the optimization dynamics; the theoretical analysis in Section 3 is developed for domains in $\mathbb{R}^3$ with smooth boundaries.

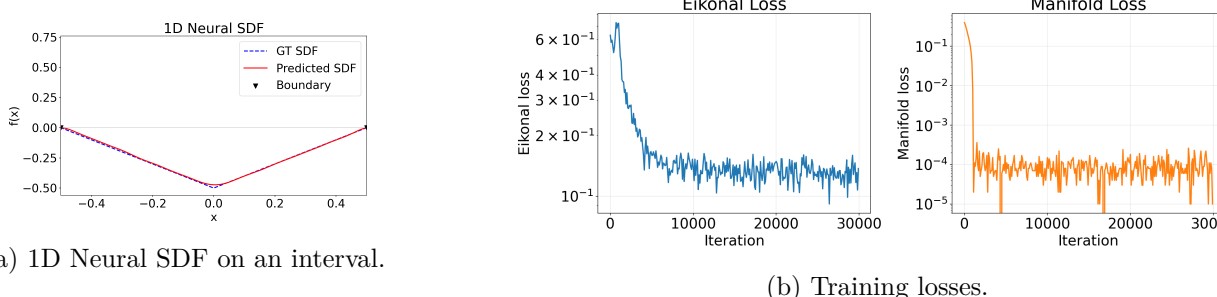

(a) 1D Neural SDF on an interval.

(b) Training losses.

Figure 12: Solution to 1D Eikonal equation. The method correctly identifies the SDF among Lipschitz solutions

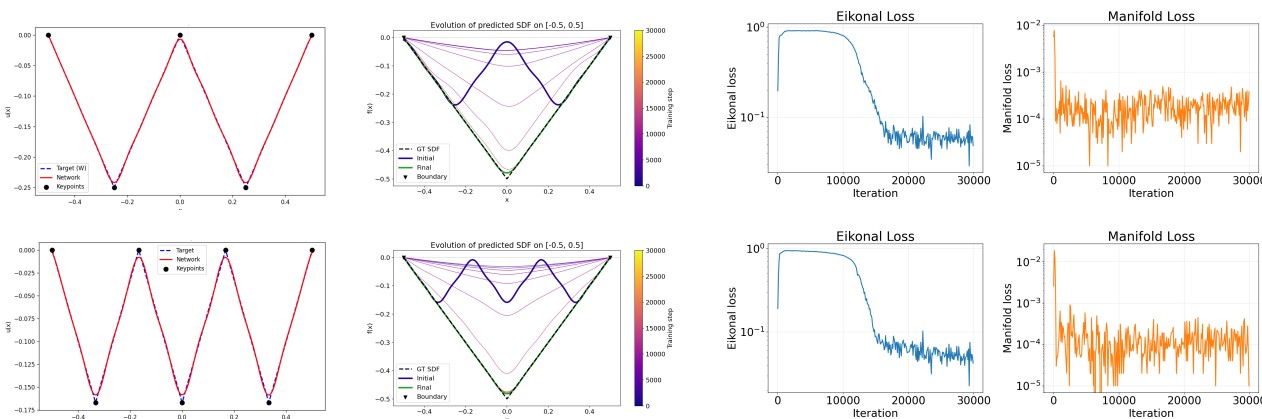

Figure 13: **Top:** W-shape initialization. **Bottom:** M-shape initialization. **Left:** Network pre-trained to fit the zig-zag target. **Center:** Evolution of the predicted SDF during training, converging from the zig-zag (purple) to the correct V-shaped SDF (green). **Right:** Eikonal and manifold loss curves.

