# OpenReview forum: "ViscoReg: Neural Signed Distance Functions via Viscosity Solutions"
_TMLR — Accepted by TMLR_

### Review · Reviewer_DnHg · 2026-02-12

**Summary Of Contributions:**

This paper revisits the SDF optimization using PDE theory, arguing that the standard practice of enforcing the Eikonal constraint with a pointwise residual loss leads to non-unique solutions and can be unstable.

The paper first frames neural SDF training as learning a solution to an ill-posed Eikonal boundary value problem with infinitely many solutions. It then derives an $L^\infty$-type generalization bound that relates the worst-case deviation from the viscosity solution to the training losses, using viscosity-solution uniqueness and stability.
Second, motivated by vanishing-viscosity theory, the authors propose ViscoReg: a viscous Eikonal loss with an annealed (decaying) viscosity coefficient. This term is intended to stabilize the gradient-flow dynamics of Eikonal training, and the authors also report better quantitative results.
The method is evaluated on multiple surface/scene reconstruction benchmarks and is reported to outperform several neural SDF baselines without significant extra compute.

**Key strengths:**
- Clear PDE-based story: Eikonal training can be non-unique, and viscosity solutions offer a principled way to pick the "right" solution.
- Simple and lightweight: ViscoReg is easy to implement (add a viscous term and anneal its weight) and appears to add little overhead.
- Broad empirical coverage (objects and scenes) with consistent reported gains over several neural SDF baselines.

**Key weaknesses:**
- The generalization bound relies on assumptions (e.g., gradients bounded away from zero, sampling/quadrature conditions) that may be hard to verify in practice.
- It is unclear how much improvement comes from the annealing schedule versus simply adding smoothing, without stronger ablations and sensitivity tests.
- The empirical claims are weakened because they do not align with many previous metric reports. More explanation required.

**Audience:**

Yes

**Audience Explanation:**

Yes, for a subset of the audience. The work connects neural SDF optimization with PDE theory and proposes a simple training regularizer; these ideas are likely to be of interest to researchers and practitioners in reconstruction and implicit modeling, although the practical implications of the theory may require further clarification.

**Broader Impact Concerns:**

I do not have major broader-impact concerns beyond standard considerations.

**Claims And Evidence:**

No

**Claims Explanation:**

- Further analysis is needed to justify how Assumption 1 and Assumption 2 can hold during neural network training. In particular, Figure 1 itself appears to provide a counterexample to Assumption 1: when ViscoReg converges, there exist many input locations where the gradient norm is zero or extremely close to zero, which contradicts the strictly positive lower bound required by Assumption 1.

- Even if we step back and assume that Assumptions 1 and 2 do hold, and thus Theorem 1 follows, it still requires a clearer explanation how this resolves the non-uniqueness issue discussed at the beginning of the Generalization Error section. As stated, Theorem 1 provides an upper bound on the $L^\infty$ norm of the difference between the learned SDF and the ideal SDF. It is not immediately clear whether such boundedness alone can imply convergence to the ideal SDF, and this point needs further analysis. Otherwise, readers may be confused about what the bound guarantees in practice: if the bound does not actually go to zero and merely remains finite, then mild zig-zag artifacts might not be eliminated by the model. If this concern is valid, the paper’s claim that the estimated solution converges to the unique viscosity solution would be weakened.

- The paper repeatedly emphasizes that the method “does not add any extra hyperparameters,” but this seems in tension with the experiments. The ablation study suggests that optimization is quite sensitive to the decaying schedule: with an inappropriate schedule, the final results are not good. This limitation needs to be discussed more explicitly, or the corresponding bullet point should be removed, or the caveat should be stated earlier. Moreover, this point does not seem unique to the proposed method; for example, StEik also does not introduce additional hyperparameters beyond a loss coefficient.

**Requested Changes:**

1. (Critical) More experiments and clearer analysis are needed to evaluate whether the method can converge to the true SDF in practice, and how informative or meaningful the bound in Theorem 1 is. For example, the authors could (i) directly assess whether the predicted values are close to the true distance values, (ii) examine whether the gradients of the predicted level sets faithfully point in the opposite direction against the surface, and (iii) include a simpler 1D sanity check. Unfortunately, based on the current Figure 1, especially when inspecting the level sets, ViscoReg appears worse than the baseline, which substantially weakens the authors' claims.

2. (Critical) The paper should further justify, with additional explanation and empirical analysis, whether the two assumptions required by Theorem 1 actually hold in the neural training setting. In particular, the zig-zag phenomenon discussed in this section can create many locations where the gradient norm is extremely close to zero.

3. (Critical) I recommend removing the claim that the method "does not add any extra hyperparameters," unless there is clearer experimental evidence showing that the proposed loss is not sensitive to the epsilon decay schedule. From the current results, it appears to be the opposite and some simple changes make the model worse than baselines. Moreover, many baselines also do not introduce additional hyperparameters beyond a loss coefficient.

4. (Strengthening) In the main text, Table 4 should not compare the with normal model in the same column as models without normals. Also, Table 1 and Table 3 should include the quad variant of StEik. Finally, adding Hausdorff distance comparisons would make the empirical evaluation more convincing.

---

> ### Author Response · Authors · 2026-04-16
> **Response to Reviewer DnHg**
>
> We thank Reviewer DnHg for their feedback. Below we address their concerns.
>
> ### Assumption 1
>
> > *"... justify how Assumption 1 ... can hold during neural network training…"*
>
> The target SDF satisfies the Eikonal ($|| \nabla u||_2 = 1$). Assumption 1 posits that the network optimization has converged to a sufficiently good local minimum where its gradient approximates that of the target and is bounded away from zero. It is a standard regularity condition to prevent singular gradient flows. If we do not assume the trained network minimizes the Eikonal loss to some reasonable degree, there is naturally no basis for convergence.
>
> We verify this empirically (Tab. 5 in the updated manuscript): evaluating the trained network on a dense 512x512 grid on the domain of the fractal shape in Fig. 1, the median gradient norm is 0.9950, and no points have gradient norm 0.
>
> | **Point Set** | **Mean** | **Median** | **Std** | **% < 0.01** | **% = 0.0** |
> | --- | --- | --- | --- | --- | --- |
> | Grid 512² |  0.9739 | 0.9950 | 0.1356 | 0.0% | 0.0% |
>
> ### On Figure 1
>
> > *"Figure 1 itself appears to provide a counterexample to Assumption 1 …"*
>
> We disagree. By the Implicit Function Theorem, the extraction of clean, continuous level sets (as seen for ViscoReg) requires that the gradient $||\nabla u||$ is non-zero on the level set. If the gradient were zero, the level set would be singular or undefined. This is also verified empirically (Tab. 5 or see above).
>
> > *"...based on the current Figure 1… ViscoReg appears worse than the baseline … "*
>
> We disagree with this assessment, and respectfully request a re-examination of the "Converged" and "Ground Truth" columns in Figure 1, specifically the reconstructed zero-level set (the solid black line) which is the surface of interest.
>
> - **SIREN (Row 1):** Fails to form a closed, watertight boundary and has incorrect interior values.
> - **DiGS (Row 2):** Boundary with self-intersections and disconnected islands.
> - **StEik (Row 3):** Exhibits multiple self-intersections.
> - **ViscoReg (Row 4):** The only method that reconstructs a smooth, connected boundary and maintains the intricate structure without self-intersections.
>
> This is confirmed quantitatively in Tab. 7 (Tab. 5 in the original manuscript), where ViscoReg's learned SDF is about 30% more accurate than SoTA and 25% better near the surface. We have added zoomed-in views in the appendix (Fig. 11) for closer inspection.
>
> ### Zig-Zag Functions
>
> > *"... the zig-zag phenomenon … create many locations where the gradient norm is extremely close to zero."*
>
> The zig-zag functions in Sec. 3 are alternate (non-viscosity) solutions to the Eikonal. As explicitly stated in the manuscript, they have slope $\pm 1$, so their gradient norm is exactly equal to one everywhere except at the measure-zero set of discontinuities. They do not have zero gradients.
>
> ### Assumption 2
>
> > *"... justify ... Assumption 2 can hold during neural network training..."*
>
> Assumption 2 is the sampling density of the input point cloud, which is a property of the dataset. It is not a property of the network or training dynamics. Assumption 2 is a standard quadrature error bound to ensure that the discrete sample sum approximates the continuous integral as $N \to \infty$, following established frameworks in generalization theory for PDE-constrained networks (e.g., Mishra & Molinaro, 2022; 2023). We explicitly state the values of $\beta$ for standard sampling techniques like uniform and Monte Carlo sampling.
>
> > *"... upper bound on the $L^\infty$ norm of the difference ... whether such boundedness alone can imply convergence... if the bound does not actually go to zero and merely remains finite.."*
>
> Convergence guarantees in numerical schemes (both classical and neural) are asymptotic. Theorem 1 guarantees convergence to the viscosity solution as the loss $\mathcal{L} \to 0$ and the number of quadrature points $\to \infty$, just as classical finite difference schemes converge as the grid is refined. A bound of exactly zero (with zero error on unseen data) is not feasible for a finite-capacity network trained on finite data.
>
> It is important not to conflate numerical approximation error with convergence to an incorrect PDE solution. Thm. 1 is derived from properties unique to the viscosity solution, no other Eikonal solution satisfies the stability estimates (Lemmas 1–2) which is used to prove it. The finite error in practice is standard network approximation error, not convergence to the wrong solution.
>
> ### Gradients Pointing in the Opposite Direction
>
> > *"...examine whether the gradients ... point in the opposite direction ..."*
>
> We point out that we have already quantitatively evaluated this using the Normal Consistency Score (NCS) in Tab. 3. NCS is the average absolute dot product between predicted gradients and ground-truth surface normals. ViscoReg achieves the best NCS (95.67), even without normal supervision, confirming our gradients align with ground-truth normals.

---

> > ### Author Response · Authors · 2026-04-16
> > **Response to Reviewer DnHg (Continued)**
> >
> > ### Annealing vs. Smoothing
> >
> > > *"... annealing schedule versus simply adding smoothing..."*
> >
> > This comparison is already central to our evaluation. The concept of "adding smoothing" is the DiGS baseline (Ben-Shabat et al., 2022), as we point out in Sec. 2.3 and 4.1. DiGS minimizes the Laplacian (mean curvature) of the field, which is smoothing. However as pointed out in our paper, this should not be done in areas of fine detail.
> >
> > -  Our experiments (Tab. 1, 2, 3) show that ViscoReg consistently outperforms DiGS across all metrics and datasets. The oversmoothing can be seen in Fig 1-4, with pronounced effects in scene reconstruction (Fig 2.).
> >
> > ### On the 1D Sanity Check
> >
> > In a 1D domain (an interval $[a,b]$), the surface boundary is not a connected manifold, but just two discrete points $\{a, b\}$. The true SDF is the piecewise linear function $u(x) = \min(|x-a|, |x-b|)$. There is no surface to extract and the data fidelity loss (manifold loss) can only be enforced at exactly two points. An overparameterized highly non-linear network with two data points to fit 2 straight lines will not train reliably. Hence we (and other baselines) rely on 2D shapes for Neural SDF sanity checks.
> >
> > ### On Hyperparameters and Sensitivity
> >
> > > *"... 'does not add any extra hyperparameters,' but this seems in tension ... sensitive to the decaying schedule... does not seem unique to the proposed method …"*
> >
> > - **Baseline Parity:** We have not claimed that the lack of extra hyperparameters is a unique advantage. Our original manuscript explicitly stated: *"Our loss does not add extra hyperparameters, **compared to** DiGS, StEik, or HotSpot"* in the introduction and later stated *"most state of the art methods (including DiGS, StEik, and HotSpot) use annealing for their loss terms, and hence ViscoReg does not add any extra hyperparameters."* Both instances frame this as parity with existing methods to convey that the improved performance does not come at the cost of additional hyperparameters over other SoTA methods. We have rephrased these sentences in the revised manuscript to remove any scope for confusion.
> >
> > - **Sensitivity:** We do not believe the ablations show our method as "quite sensitive." Tab. 2 shows that whether we use fast decay ($d_C=0.19$, $d_H=3.51$), slow decay ($d_C=0.18$, $d_H=3.28$), quintic ($d_C=0.19$, $d_H=3.89$), or double the baseline ($d_C=0.18$, $d_H=3.17$), the performance is stable and comparable to SOTA best ($d_C=0.19$, $d_H=3.17$). The method is fairly robust to the schedule; it only fails if viscosity is removed entirely which is the SIREN method. We show additional ablation on the decay schedule for real scans in Tab. 16 (Tab. 14 in the original manuscript) of the appendix, where we again show robust performance for a range of decay schedules.
> >
> > ### Alignment of Metric Reports
> >
> > > *"... do not align with many previous metric reports."*
> >
> > Most metrics reported in our paper are taken from previously published results (see Ben-Shabat et al., 2022; Yang et al., 2023). As detailed in Sec 5., for an apples-to-apples comparison, we evaluate methods on the standard linear architecture (4 layers, 256 channels). StEik uses quadratic layers for their results and HotSpot claimed in their paper to have used a smaller 5/128 architecture which is much smaller than our 4/256 model and yielded worse results on testing. Our original manuscript made explicit notes about these choices in the Sec. 5 with details in the Appendix.
> >
> > ### Other Changes (Tables and Hausdorff)
> >
> > - We have updated Tables 1 and 3 to include the quadratic variant of StEik (StEik quad).
> >
> >   **Table 1 (updated rows):**
> >
> >   | Method | $d_C \downarrow$ | $d_H \downarrow$ |
> >   | --- | --- | --- |
> >   | StEik (linear) | 0.20 | 4.56 |
> >   | Ours | **0.18** | **2.76** |
> >   | StEik (quad) | **0.18** | 2.80 |
> >   | Ours (quad) | **0.18** | **2.69** |
> >
> >   **Table 3 (updated rows):**
> >
> >   | Method | $d_C \times 10^{-2} \downarrow$ | F-score $\uparrow$ | NCS $\uparrow$ |
> >   | --- | --- | --- | --- |
> >   | StEik (lin) | 38.45 | 86.31 | 94.73 |
> >   | StEik (quad) | 39.56 | 82.62 | 89.82 |
> >   | Ours (lin) | **30.55** | 86.50 | **95.67** |
> >
> > - **Table 4:** We have clarified Tab. 4 caption to explicitly note that the first group of methods above the horizontal line use normal supervision, while those below do not. Several recent normal-free methods (including ViscoReg, HotSpot, and StEik) outperform normal-supervised methods. Thank you for pointing this out.
> >
> > - **Hausdorff distances:** Tab. 1 provides Hausdorff metrics for the SRB dataset. For real scans, we evaluate on the standard suite of metrics (Chamfer distance, F-score, and NCS) provided by the benchmark authors (Huang et al., 2024). For ShapeNet, we report the same standard metrics (Squared Chamfer and IoU) utilized by the primary baselines we compare against (DiGS and StEik), ensuring consistent literature comparisons.

---

> > ### Comment · Reviewer_DnHg · 2026-04-17
> > **zig-zag phenomenon**
> >
> > Thanks for your explanation.
> >
> > Thank you also for including the Normal Consistency Score experiments. They address some of my concerns, but an important issue remains.
> >
> > As you acknowledged, I would suggest replacing Ω with Ω \ N in many places in the paper, where N is a measure zero set. Once this correction is made, several parts of the analysis may be materially affected.
> >
> > This is why, even if all the theorems are correct as stated, I still do not see a direct answer to the main problem posed by the paper: With infinitely many solutions to the Eikonal equation, why is minimizing the PDE residual loss on a finite training set sufficient to ensure convergence to the unique viscosity solution (i.e., the SDF)? Specifically, how could you make sure the viscosity solution becomes the SDF when the domain is corrupted by a measure zero set?
> >
> > My concern is that the zig-zag phenomenon may persist, and the proposed method may not be able to escape such a local optimum. Even if one densifies the samples or increases the network capacity to reduce the training error, the learned function may still converge to a zig zag solution. Empirical evidence would therefore be very helpful, especially a toy example that starts from a W shaped function and successfully optimizes to a V shaped function.
> >
> > For example, suppose the target SDF is
> > target(x) = |x| - 2
> > with boundary points (-2, 0) and (2, 0). Why would the method not converge instead to
> > fake1(x) = ||x| - 0.5| - 1.5
> > or
> > fake2(x) = ||x| - 1.0| - 1.0?
> > Although you state that “no other Eikonal solution satisfies the stability estimates (Lemmas 1–2),” I am still not fully convinced you can distinguish them in practice.
> >
> > In my view, the viscosity solution can distinguish the target function from these fake solutions only if the method can clearly distinguish the domain and the boundary condition in Eq. (7). In practice, if the boundary is given only by the two points (-2, 0) and (2, 0), one may unintentionally introduce measure zero points at which the Eikonal equation (1) does not hold. More generally, if the domain is not properly specified, it seems possible to construct a sequence of viscosity solutions that converges to such fake functions.
> >
> > I therefore think this issue needs to be addressed much more carefully in the discussion. In particular, I would like to see a concrete one dimensional example showing how fake1(x)/fake2(x) can be optimized into target(x), together with the corresponding loss curves.
> >
> > This is my main concern, and essentially my only remaining one.

---

> > > ### Author Response · Authors · 2026-04-17
> > > **Resonse to zig-zag phenomenon**
> > >
> > > We thank the reviewer for their reply and for noting that they essentially have only one concern left. We want to address each point carefully. To confirm, the viscosity solution is the name given to the unique physically meaningful solution of the Eikonal, i.e., the SDF. It is obtained in the limit of solutions to well-posed parabolic equations.
> > >
> > > **THEORY**
> > >
> > > > *"I would suggest replacing $\Omega$ with $\Omega \setminus \mathcal{N}$... several parts of the analysis may be materially affected."*
> > >
> > > To clarify what was acknowledged in our previous response: we stated that the zig-zag functions have slope $\pm 1$ everywhere except at a measure-zero set of $C^1$ discontinuities. This is just to say they are Lipschitz (almost everywhere) solutions of the Eikonal, not an acknowledgment of any issue with the domain $\Omega$.
> > >
> > > Equation 1 is a boundary value problem whose solution is sought in the viscosity sense. This framework was introduced to handle equations like the Eikonal where classical solutions do not exist. The viscosity solution (Definition 3.1; Definition A.1 in the original manuscript) is defined through sub/supersolution conditions using smooth test functions $\phi$. The gradient $\nabla \phi$ of the test function is evaluated, we never evaluate $\nabla u$. The measure-zero set where $u$ is non-differentiable therefore does not enter the definition, the stability estimates (Lemmas 1–2), or Thm. 1. Integrals over measure-zero sets are also zero. The domain $\Omega$ requires no modification, and no part of the analysis is affected.
> > >
> > > Zig-zag functions, while satisfying $|u'| = 1$ a.e., fail the viscosity sub/supersolution test at their kink points and hence are **not viscosity solutions** per Definition 3.1.
> > >
> > > For these reasons, the standard mathematical formulation uses $\Omega$ without excluding the measure-zero set $\mathcal{N}$.The viscosity formulation is already well-defined on $\Omega$ without excluding any set. This is standard in the literature and consistent with prior Neural SDF works. However, to make this distinction explicit, we now refer to the other solutions as "Lipschitz solutions" (almost-everywhere solutions: solutions defined everywhere outside a set of measure zero) throughout the revised manuscript.
> > >
> > > > *"how could you make sure the viscosity solution becomes the SDF when the domain is corrupted by a measure zero set?"*
> > >
> > > The viscosity solution of Eq. 1 with boundary $\partial\Omega$ is the SDF, this is a classical result (Thm. 2) not something our method needs to enforce. This holds by the definition of viscosity solutions and does not depend on measure-zero sets.
> > >
> > > > *"the viscosity solution can distinguish the target function from these fake solutions only if the method can clearly distinguish the domain and the boundary condition"*
> > >
> > > The viscosity framework distinguishes them at every point through the sub/supersolution conditions in Definition 3.1, not through the domain specification. A zig-zag fails the subsolution test at its kinks regardless of how the domain is specified.
> > >
> > > > *"it seems possible to construct a sequence of viscosity solutions that converges to such fake functions"*
> > >
> > > The viscosity solution of a given BVP is unique (Lemma 1). There is no sequence,  there is exactly one. If the reviewer means a sequence of solutions to the parabolic equations,, these converge uniformly to the viscosity solution/SDF (Thm 2), not to zig-zag functions. This was proven by Crandall & Lions (1983).
> > >
> > > **EXPERIMENTS**
> > >
> > > > *"Why would the method not converge instead to fake1(x) or fake2(x)? ... I am still not fully convinced you can distinguish them in practice … I would like to see a concrete one dimensional example showing how fake1(x)/fake2(x) can be optimized into target(x)"*
> > >
> > > fake1 and fake2 are Lipschitz solutions of the Eikonal, they satisfy $|u'| = 1$ a.e. Theoretically, Theorem 1 guarantees that the neural SDF converges to the SDF and not any other almost-everywhere solutions.
> > >
> > > Per our understanding of this request, to show empirically that the network does not converge to local minima of zig-zag functions, we reconstruct the 1D SDF using a small SIREN network with the SDF data values given only at the boundary points as usual. The network converges to the correct V-shaped SDF and does not converge to a zig-zag (see Fig. 12 in the revised appendix, with the SDF and the Eikonal and manifold losses). This is to be expected from Thm. 1 and also noting that the Eikonal loss will be higher for functions with more kinks when using smooth activation functions (as we do with the SIREN architecture) to approximate non-smooth targets. We note that the qualitative results presented in Fig. 1-4 also indicate that in 2D and 3D too, the network seems to converge to the SDF and not other Lipschitz solutions.
> > >
> > > We hope we were able to satisfactorily answer your questions. Please let us know if there are any further concerns.

---

> ### Comment · Reviewer_DnHg · 2026-04-17
> **On ESCAPING Zig-Zag Local Minima**
>
> My concern still remains. In your paper, you mention local minima multiple times, and at the mathematical level you also formulate a unique target. Naturally, this gives the reader the impression that your method is able to escape local minima and converge to that uniquely correct SDF. Therefore, what I would like to see now is an experiment showing what happens when the model is not well initialized and has already fallen into the kind of local minima that other methods may get stuck in, namely the zig-zag solutions such as the fake1 and fake2 examples I mentioned. How, then, does your method escape from them?
>
> This experiment would require first initializing the model at fake1 and fake2, which should not be difficult. You could even do this directly through pointwise function value constraints. Then, with the same boundary conditions, namely (±2,0), you start training using your method. I would like to observe both the evolution of the loss and the evolution of the function plot, until the model eventually escapes the local minimum and converges to the desired V-shaped SDF. Only such an experiment would support your claim.
>
> This seems well motivated to me. A small zig-zag configuration could plausibly arise from an imperfect initialization or from stochastic perturbations during optimization, so such a local optimum could realistically occur in practice. That is why I would like to see an explicit escape trajectory. An experiment from a relatively smooth zig-zag initialization like fake3(x) = cos($\pi$x) - 1 fake4(x) = cos(2$\pi$x) - 1 would also be acceptable.

---

> > ### Author Response · Authors · 2026-04-20
> > **Response to Escaping Zig-Zag Local Minima**
> >
> > Thank you for the response. We note that the experiment in our previous response (training from standard initialization and showing the network converges to the correct SDF, not to fake1 or fake2) was a direct answer to the reviewer's earlier question: *"Why would the method not converge instead to fake1(x) = ||x| - 0.5| - 1.5 or fake2(x) = ||x| - 1.0| - 1.0?"* We address below a different question: whether the network can escape if initialized at a zig-zag.
> >
> > > *"your method is able to escape local minima and converge to that uniquely correct SDF"*
> >
> > We wish to clarify a confusion regarding what our paper claims and does not claim.
> >
> > - First, ViscoReg is introduced to stabilize the gradient flow of the Eikonal loss, which potentially can help avoid poor local minima in practice, as evidenced by the improved performance in Sec. 5. We do not claim it escapes any arbitrary local minima.
> >
> > - Second, Theorem 1 states that the $L^\infty$ deviation from the viscosity solution is bounded by the training errors. As $\mathcal{L} \to 0$ and $N \to \infty$, the network converges to the viscosity solution. If the training error remains high (for instance, if the network is stuck at a local minimum) the bound is loose, and we make no guarantee that the network will escape such a configuration. Escaping local minima clearly depends on the choice of optimizer and training schedule.
> >
> > That said, per the reviewer's request, we initialized the network at zig-zag functions and trained the Neural SDF. The network does escape the zig-zag and converges to the correct SDF (see Figure 13 in the revised appendix). We note that Theorem 1 is developed for domains in $\mathbb{R}^3$ with smooth boundaries; the 1D experiment is provided as an empirical demonstration.

---

> > > ### Comment · Reviewer_DnHg · 2026-04-21
> > > **Discussion about zig-zag.**
> > >
> > > Thank you for the detailed responses. The discussion has clarified several of my earlier concerns, especially regarding the intended scope of the theoretical claim. In particular, I appreciate the clarification that the paper does not claim escape from arbitrary local minima in full generality, and I find the added zig-zag initialization experiment useful. At this point, my view is that if the final version addresses the points below more explicitly, then I would support a Leaning Accept recommendation.
> > >
> > > The main issues I would still like to see addressed are:
> > >
> > > 1. The manuscript should further moderate its practical claims about escaping local optima. I find the current response more appropriately phrased than some parts of the paper itself: wording such as “potentially” is scientifically more accurate here. As written, the paper still contains several statements whose tone is too strong and may leave the reader with the impression that the method reliably escapes such local optima, whereas Figure 1 already suggests that this is not always the case, at least in some far regions with gradient norm zero which shouldn't be there.
> > >
> > > You can say they have no measure, but in this case, this error also influences the accuracy for distance queries outside. Regardless of whether this is ultimately due to numerical precision or some other practical limitation, the manuscript should use more careful qualifiers, for example “theoretically,” “partially,” or similar wording where appropriate.
> > >
> > > 2. The paper should describe more concretely what is observed when the method does escape a zig-zag local optimum. For example, does the function first become smoother before converging to the target SDF? Based on the current presentation, the choice of the annealed epsilon schedule may be important in this process, and if the escape behavior is tied to the annealing phase, this relationship should be analyzed more directly.
> > >
> > > It is also worth noting that this behavior may depend on the optimizer dynamics: in the latest Fig. 13, both losses appear to increase during the first few iterations. This suggests that the loss design does not simply imply a monotone loss decrease followed by convergence to the unique target SDF, and this point should be discussed more carefully.
> > >
> > > 3. Beyond wording, the limitations should then be stated more clearly. In Figure 1, zig-zag-like behavior still appears to remain in regions far from the center, and the paper should discuss whether this is mainly due to finite optimization accuracy, limited sample density, or limited model capacity, or whether it indicates that some local optima may persist in practice. More generally, the paper should better characterize which types of zig-zag or bad local optima appear empirically escapable, and which types may still trap the method in practice. This would make the practical scope of the method more precise.

---

> > > > ### Author Response · Authors · 2026-04-24
> > > >
> > > > Thank you for the constructive feedback. We have made the following changes to the revised manuscript:
> > > >
> > > > - We have added qualifiers in the abstract, introduction and following sections (e.g., "can be theoretically proven to stabilize" rather than "provably stabilizes", “guarantees” to “theoretical guarantees”).
> > > > - We have expanded the limitations section (Sec. 6) and explicitly note that while ViscoReg stabilizes the gradient flow, escape from arbitrary local minima is not guaranteed and depends on the optimizer, initialization, and training schedule.
> > > > - We have added discussion of the escape trajectory in A.5.12, including the non-monotone loss behavior (which may be expected with multiple loss terms, where reducing one term may temporarily increase another).
> > > >
> > > >
> > > > Regarding the scope of the paper: our contributions are a generalization bound (Theorem 1) and a regularizer with theoretically provable stable gradient flow (Sec. 4.2). While we have expanded on the practical limitations, a detailed study of which local minima neural SDFs can or cannot escape is an important question in nonconvex optimization that we believe merits its own dedicated investigation. We have acknowledged this in the limitations.
> > > >
> > > >
> > > > We have carefully reviewed the manuscript and if there are specific sentences where the language feels too strong in the revised manuscript, we would appreciate you pointing them out, and we would be happy to modify it.

---

### Review · Reviewer_4jEC · 2026-02-16

**Summary Of Contributions:**

- Derive an $L^\infty$ generalization bound in terms of boundary and $L_1$ PDE residuals at discrete collocation points.
- Introduce a viscosity-based regularization inspired by vanishing-viscosity theory for Hamilton--Jacobi equations.
- Show significant improvements on multiple surface reconstruction benchmarks.

**Audience:**

Yes

**Audience Explanation:**

Yes. The paper connects PDE theory with neural implicit surface reconstruction and provides both theoretical analysis and empirical gains. This should be of interest to researchers working on geometric deep learning, implicit neural representations, and physics-informed machine learning.

**Broader Impact Concerns:**

No concern

**Claims And Evidence:**

Yes

**Claims Explanation:**

The main claims are supported by both theory and experiments. The paper derives an $L^\infty$ bound relating reconstruction error to boundary and PDE residuals (PINNs style), and the proposed viscosity-based regularization is motivated from Hamilton-Jacobi theory. Empirically, the method consistently improves reconstruction metrics across multiple benchmarks.

`

**Requested Changes:**

**General comments.**
The paper is very well written. The idea is explained clearly, and the results (at least from someone not deeply involved in scene reconstruction) look genuinely strong and quite impressive.

**Missing theoretical bridge (main concern).**
I understand the stability result for the inviscid Eikonal equation, and I understand from Hamilton–Jacobi theory that $u_\varepsilon \to u$ as $\varepsilon \to 0$. However, why the model which minimizes the viscous empirical loss would approximate $u_\varepsilon$? And consequently, why does annealing $\varepsilon$ during training implement vanishing viscosity at the PDE level? Conceptually it makes sense, but this bridge is not made explicit. Perhaps it would make sense to derive the stability bounds for the viscous equation as well.

**Ablation on sampling / bound dependence.**
The bound explicitly depends on the number of sampled points, but there is no experiment varying the sampling density. It would be very interesting to see how reconstruction quality and stability behave as the number of surface/domain samples changes.

**Compute cost / second-derivative overhead.**
Since the method requires Laplacians, the computational overhead is non-negligible compared to first-order methods. It would be nice  to see a clearer comparison in terms of runtime and memory footprint relative to DiGS/StEik/HotSpot. Especially if the method is to be used in larger-scale settings, this matters.

**On the bound vs.\ practical metrics.**
The bound is formulated in $L^\infty$, whereas the evaluation is done with Chamfer distance and IoU. A short discussion clarifying how (qualitatively) the bound relates to these metrics would make the theory feel more connected to the empirical section.

---

> ### Author Response · Authors · 2026-04-16
> **Resonse to Reviewer 4jEC**
>
> We thank the reviewer for their constructive feedback! Below we address their concerns.
>
> ### Missing Theoretical Bridge
>
> Thank you for this insightful question! As you note, vanishing viscosity serves as the motivation for our loss. We do not claim that the network approximates intermediate solutions $u_\epsilon$ at each step, nor that annealing perfectly matches vanishing viscosity at the strict PDE level; this would be quite hard to show. The initial viscosity helps to stabilize the gradient flow of the Eikonal loss and this can be proven theoretically, as we do in Section 4.2 and A.3/A.4.
>
> Since $\epsilon$ is annealed completely to zero well before training ends, the final phase of training minimizes the standard, inviscid Eikonal loss. Thus, regardless of the intermediate training trajectory, the final converged state is governed entirely by Theorem 1, which bounds the error of our final solution by the Eikonal and boundary errors.
>
> Regarding the suggestion to derive "stability bounds for the viscous equation as well": We have already provided these stability estimates for the gradient flow of the viscous equation in Section 4.2 and Appendix A.4, showing the viscous formulation is unconditionally stable, unlike the inviscid Eikonal. If the reviewer instead meant an *error* bound for the viscous equation: as stated above, Theorem 1 covers our converged method. We have included a few clarifying sentences in the revised manuscript to make this bridge explicit.
>
> ### Ablation on Sampling / Bound Dependence
>
> Per your request, we have performed a new ablation study on the Surface Reconstruction Benchmark (SRB), limiting the number of ground-truth manifold points to 10%, 40%, 60%, and 80% of the full point cloud.
>
> As expected, there is a degradation in reconstruction quality as the number of manifold points decreases. However, sampling for INR is an interesting topic that has entire papers devoted to it (e.g., Pais et al. [2024], Lin et al. [2024]), and a deeper study may be beyond the scope of our current work. In our framework, we just assume that the chosen sampling technique satisfies Assumption 2 to ensure the discrete sum adequately approximates the true integral.
>
> We do agree this deserves its own treatment. We have added Tab. 6 and the above references to the revised manuscript for the interested reader to dive deeper.
>
> | **Manifold Points** | **Chamfer Distance** | **Hausdorff Distance** |
> | --- | --- | --- |
> | **10%** | 0.292 | 4.37 |
> | **40%** | 0.227 | 4.36 |
> | **60%** | 0.198 | 3.71 |
> | **80%** | 0.194 | 3.66 |
> | **100%** (Baseline) | **0.176** | **2.76** |
>
> ### Compute Cost / Second-Derivative Overhead
>
> Our original manuscript provided a timing analysis with respect to other second-order methods (DiGS and StEik) in the appendix. We have expanded it to include HotSpot as well (Tab. 8 in original, Tab. 10 in revised). ViscoReg evaluates at 35.50 ms, which is actually slightly faster than our direct second-order competitor DiGS (36.05 ms), while purely first-order methods like HotSpot are naturally faster (~20 ms).
>
> However this trade-off may be necessary for more accurate reconstruction (as seen by the improved performance of our method for SRB, Real scans and Figures 3–4). In many applications like medical imaging and physics simulations, high-fidelity surface reconstruction is an offline process with ample computational resources. In these pipelines, the structural integrity of the final watertight mesh is more important, and can outweigh slight computational overhead.
>
> ### On the Bound vs. Practical Metrics
>
> - Theorem 1 bounds the $L^\infty$ norm, which measures the worst-case distance between two functions (ground-truth and predicted SDF) over the whole domain.
>
> - In surface reconstruction, the metric of interest is the quality of the extracted mesh of the zero level set compared to a ground-truth mesh. This is primarily evaluated using metrics such as Chamfer distance, which calculates the average nearest-neighbor distance between points on these two discrete surfaces, or Intersection over Union.
>
> While these measure different things, the error of the network, of course, directly controls the quality of the extracted mesh. If the network has local volumetric patches with high $L^\infty$, the zero-level set of this network can manifest the errors as "ghost geometries" or holes. A mesh with these ghost geometries will have a high Chamfer error or low IoU. So, while there is no direct formula linking the two, we can see that a small $L^{\infty}$ error ensures good mesh quality metrics. We have added a brief discussion clarifying this relationship in the revised text.
>
> Lin et al. "On Optimal Sampling for Learning SDF Using MLPs Equipped With Positional Encoding." *IEEE Transactions on Visualization and Computer Graphics,* 2024.
>
> Pais et al. "A Probability-Guided Sampler for Neural Implicit Surface Rendering." *ECCV* 2024.

---

> > ### Comment · Reviewer_4jEC · 2026-04-24
> >
> > Thank you very much for addressing my comments.

---

### Review · Reviewer_915N · 2026-04-12

**Summary Of Contributions:**

Summary

This paper proposes to use viscosity solutions of Eikonal equation as the signed distance function.
Using the fact that viscosity solution can be uniquely determined and distance between the viscosity solutions of different Eikonal equations can be upper bounded by the difference in the conditions of Eikonal equatios, the author theoretically analyzed the generalization error of the obtained function.
Further, the author proposes to add small diffusion term to Eikonal equation to make the problem well-posed and whose solution converges to viscosity solution of original equation as the diffusion term goes to 0.
The author theoretically discusses the stability of the proposed method and experimentally demonstrates that the proposed method demonstrates good surface reconstruction quality on both synthetic and real data.


Strength

The proposed method is well theoretically-motivated and experimentally works well.

The author conducts several experimental evaluations on both synthesized and real data. The proposed method demonstrates better accuracy than existing signed distance function training method.

The author conducts several ablation study and visualization to demonstrate the effectiveness of the proposed


Weakness

I think definition and property of viscosity solution would be included in the main paper rather than supplementary materials since this is closely related to the main contribution.
Further, I want to see more discussion on the motivation to use viscosity solution. Is it introduced for the sake of optimization, or is the solution itself good for modeling the shape?

Similar to Figure 1, I want to see the visualization of solution for parabolic equation with several $\epsilon$ to see the convergence of viscosity solution rather than convergence of training.

I have some questions about Theorem 1.
Though both $u_\theta*$ and $\hat{u_\theta*}$ share the same boundary condition $g_\theta*$, it need not 0. Can we apply lemma 2 in this case? If so, I think it would be better to modify the assumption of Lemma 2.

Further, if my understanding is correct, $g_\theta*$ and $f_\theta*$ are defined on all the space and the integral is approximated by sampled sum in Eq. (31). But the model $u_\theta*$ itself is trained to minimize the sampled sum. Therefore, I think some analysis of overfitting like VC theory is required.

Further, since the Theorem 1 does not use non-manifold term $L_{nm}$, it would be better to discuss the motivation to introduce this term.

**Audience:**

Yes

**Audience Explanation:**

The proposed method and theoretical evaluation will be interesting for those who work on 3d shape reconstruction.

**Broader Impact Concerns:**

The author discussed some potential concerns in Broader Impact section after conclusion.

**Claims And Evidence:**

Yes

**Claims Explanation:**

The proposed method is well theoretically-motivated and the author both theoretically and experimentally demonstrates the effectiveness of the proposed method.

I have some question about theoretical derivation in weakness section.

**Requested Changes:**

More discussion about motivation of using viscosity solution.

Visualization of solution of intermediate solution of parabolic equation.

Clarification about question on theoretical part.

---

> ### Author Response · Authors · 2026-04-16
> **Response to Reviewer 915N**
>
> We thank the reviewer for their helpful feedback. Below we address their concerns.
>
> ### More Discussion about Motivation of Using Viscosity Solution
>
> > *"...definition and property … included in the main paper..."*
>
> We agree that introducing the definition and limit property of the viscosity solution earlier improves the readability of the manuscript, and we have moved this section to the main paper in the revised text.
>
> > *"Is it introduced for the sake of optimization, or is the solution itself good for modeling the shape?"*
>
> Both. As we point out in the introduction and in Sec 3, the SDF is the unique viscosity solution of the Eikonal, so the level set of the viscosity solutions is the exact shape/surface we want to recover. No other Eikonal solution is meaningful in this context. Simultaneously, the vanishing viscosity method provides motivation for the ViscoReg loss which has a provably stable gradient flow which helps optimization.
>
> We have added more to the discussion in Sec 1, 3, 4 to make this clearer.
>
> ### Visualization of Solution of Intermediate Solution of Parabolic Equation
>
> > *"... want to see the visualization of solution for parabolic equation with several epsilon to see the convergence of viscosity solution rather than convergence of training."*
>
> To show the convergence of the viscous parabolic solution to the viscosity solution, we have added Figure 5 to the revised manuscript appendix. For this illustration, we use the closely related squared Eikonal equation ($|| \nabla u||^2_2 = 1$) and its viscous counterpart which shares the same vanishing-viscosity limit (the true SDF) as the standard Eikonal but, unlike the latter, admits closed-form solutions via the Cole–Hopf transformation.
>
> We note that illustrating this convergence through neural networks (i.e., training a separate network for each fixed $\epsilon$) would be confounded by neural optimization effects (e.g., the $\epsilon = 0$ SIREN case produces poor results for the SDF, as shown in Fig. 1 and in Tab. 1,3,4).
>
> ### Clarification about Question on Theoretical Part
>
> > *"... share the same boundary condition, it need not 0. Can we apply lemma 2 in this case?"*
>
> Thank you for pointing this out to us! To address it, we have made a slight adjustment in the proof. The auxiliary function $\hat u_{\theta^\ast}$ is now defined as the unique viscosity solution with slowness $f_{\theta^{\ast}}$ and a zero boundary condition ($\hat{u}_{\theta^{\ast}} = 0$ on $\partial\Omega$). By making this small adjustment and applying Lemma 2 (with boundary condition 0) and Lemma 1 respectively to the triangle inequality split in Eq. 31, we arrive at the exact same bounds as before. Consequently, the final conclusion of Theorem 1 follows identically, and the original statement of Lemma 2 stands as written.
>
> > *"... the integral is approximated by sampled sum … some analysis of overfitting like VC theory is required."*
>
> To obtain generalization error estimates, VC theory may be applied, but we would like to respectfully clarify that our analysis is self-contained and complete without VC theory.
>
> **VC Theory:** This framework bounds generalization by measuring the complexity of the hypothesis class. We did not adopt this approach because for deep, overparameterized networks (like Neural SDF), VC dimension scales with the parameter count and yields loose bounds. Furthermore, as the Eikonal loss evaluates the gradients of the network, applying VC theory would require bounding the statistical complexity of the network's derivative class, which could get quite intractable.
>
> **Numerical Analysis Setting (Our Approach):** The established framework for PDE-constrained neural networks (e.g., Mishra & Molinaro, 2022; Shin et al., 2020) instead bounds continuous error via numerical integration. VC theory evaluates the risk of overfitting to an unknown statistical data distribution. We treat the sampled spatial points as quadrature nodes, not as a statistical dataset. Consequently, the "overfitting" gap between the continuous integral and the discrete sum is a numerical approximation error. We explicitly formalize and bound this gap via Assumption 2. Because this discrete-to-continuous gap is already mathematically accounted for and in Theorem 1, an additional statistical learning bound like VC theory is not necessary.
>
> > *"... non-manifold term, … discuss the motivation to introduce this term."*
>
> To clarify, we do not introduce the non-manifold loss term. This is a loss term used in the original SIREN (Sitzmann et al., 2020) to improve results, and is theoretically not a necessary component. However, oscillations right at the zero-level set can cause mesh extraction algorithms (like Marching Cubes) to output wrong geometry. The $\mathcal{L}_{nm}$ term was introduced by SIREN as a standard local topological prior to suppress artifacts, and improve practical mesh extraction.

---

> > ### Comment · Reviewer_915N · 2026-04-21
> >
> > Thank you for the clarification. The modification of $\hat{u_\theta}$ looks fine.
> >
> > As for assumption 2, does it assume "for any model parameters $\theta$, $g_\theta$ satisfies the error bound inequality" or does it consider some specific $g$?
> >
> > Further, do we also require assumption about $f_\theta$ to bound the second term of Eq. (33)?

---

> > > ### Author Response · Authors · 2026-04-24
> > >
> > > Thank you for the question!
> > >
> > >
> > > Assumption 2 is a condition on the point cloud sampling, requiring that the discrete sum adequately approximates the continuous integral. The rate $\beta$ is determined by the sampling strategy (e.g., $\beta = 1/2$ for Monte Carlo). The constant $C_g$ is finite for any smooth $g$ in bounded domains, and since SIREN networks are smooth, this is satisfied for $g_{\theta^{\ast}}$.
> > >
> > >
> > >
> > > Regarding $f_{\theta^{\ast}}$: yes, a similar quadrature bound is needed. The domain points are uniformly sampled, so their quadrature error is controlled directly in the proof. Assumption 2 is stated separately for the boundary points because these come from sensor data and may have non-uniform or irregular sampling that must be accounted for independently.

---

> > > > ### Comment · Reviewer_915N · 2026-04-24
> > > >
> > > > Thank you for the answer.
> > > > Does coefficient $C_g$ depend on $g$?
> > > > Since $g_{\theta*}$ varies according to sampling, I think we may need uniform upper bound of $C_g$ i.e. $\exists C  \mathrm{s.t.}  C \geqq C_{g_\theta} \mathrm{for} \forall \theta \in \Theta$.

---

> > > > > ### Author Response · Authors · 2026-04-24
> > > > >
> > > > > Thank you for the insightful question! We wish to clarify how Assumption 2 comes up in Theorem 1. At the end of training, we have a given fixed network $u_{\theta^*}$ trained on a collection of points. We want to quantify its distance from the true viscosity solution using the error evaluated at a finite set of points. For this, we need to know how well the sample sum approximates the continuous integral for this fixed function. Assumption 2 provides this. While the training process determines which $\theta^\ast$ we arrive at (and hence which $g_{\theta^\ast}$​ we obtain the generalization bound for), this dependence is not needed for the analysis. Theorem 1 takes $\theta^\ast$ as the converged network parameter and bounds the error of that specific network. We follow a similar setting as in the works of Mishra and Molinaro 2022,2023.
> > > > >
> > > > > The constant $C_g$ depends on the regularity of this given fixed function, which is finite since this SIREN network with bounded weights has bounded $C^k$ norms. The uniform bound mentioned in the reviewer’s comment is thus a stronger assumption than what we need.

---

### Decision · Action_Editor_4okU · 2026-05-14

**Recommendation:** Accept as is

**Audience:**

Yes

**Audience Explanation:**

The work is interesting to researchers who work in 3D reconstruction.

**Claims And Evidence:**

Yes

**Claims Explanation:**

The paper provides a rigorous, PDE-motivated perspective on neural SDF optimization and validates the effectiveness of the resulting regularizer in comprehensive experiments.